**communications** engineering

# Uncovering drone intentions using control physics informed machine learning
Adolfo Perrusquía [1] ✉, Weisi Guo[1], Benjamin Fraser[1] & Zhuangkun Wei[1]

Unmanned Autonomous Vehicle (UAV) or drones are increasingly used across diverse application areas. Uncooperative drones do not announce their identity/flight plans and can pose a potential risk to critical infrastructures. Understanding drone's intention is important to assigning risk and executing countermeasures. Intentions are often intangible and unobservable, and a variety of tangible intention classes are often inferred as a proxy. However, inference of drone intention classes using observational data alone is inherently unreliable due to observational and learning bias. Here, we developed a control-physics informed machine learning (CPhy-ML) that can robustly infer across intention classes. The CPhy-ML couples the representation power of deep learning with the conservation laws of aerospace models to reduce bias and instability. The CPhy-ML achieves a 48.28% performance improvement over traditional trajectory prediction methods. The reward inference results outperforms conventional inverse reinforcement learning approaches, decreasing the root mean squared spectral norm error from 3.3747 to 0.3229.

Proliferation of cheaper drone technology has magnified the threat space for autonomous platform attacks on critical national infrastructure, defence, and national security facilities[1]. Representative examples include both intended and unintended malicious activities derived by pilot errors, incompetence, and misuse of drones. Protection against malicious drones is critical to ensuring smooth operation of services, whilst safeguarding it against the most severe threats. The fundamental research problem is that drone intention is a hidden attribute that cannot be observed directly from any perception or detection system[2,3] and, in consequence, it makes difficult to determine whether a malicious intention is or will be carried out. This creates either too many false positives (e.g., constantly suspecting anomalies) or over trusting autonomous systems.

Many efforts have been made to classify intention from observational data using experts-knowledge methods[4]. These methods define low-dimensional behavioural features[5] for relatively simple motion dynamics based on either geofence planning methods[6,7] and expert traffic rules[8] or drone's flight constraints[9]. However, the challenge of predicting intention is exasperated by an inherent cognitive bias problem caused by the low scalability of these simplistic features to complex and diverse classification of drone intention. In contrast with intention classification approaches[10,11], intention inference methods have been adopted to predict the future trajectory of autonomous systems[12] and pedestrians[13]. The majority of these methods are data-driven learning models that use snap-shot data to cluster together drone attributes[7] to predict the trajectory in several time steps in the

future[14,15]. However, the continuous flight physics has been left aside despite providing crucial information about the mission profile and intention. Physics informed models have demonstrated improvements in the learning capabilities of data-driven methods[16]. These physics informed models appear either as a regularization term in the loss function[17] or from conservation laws[18] and a prior model structure[19]. Although a great effort has been made to detect behavioural anomalies, there is still a gap in uncovering the hidden nature of intention and the associated complex capability of drones.

Here, a CPhy-ML framework is developed to uncover the hidden intention of drones without providing explicit behavioural features to the model architecture. This is done by combining the complementary merits[20] of data-driven methods with flight physics and control to regularise and stabilize the learning manifold, whilst maintaining the dynamic properties of the mission profile. This allows one to infer drone's intention by evaluating the connection between the drone's purpose of use and its observed mission profile and to increase the confidence of the predictions. One way of looking at this is to attribute intention to a family of similar mission profiles with similar high-dimensional features. In addition, the incorporation of control measurements give an additional degree-of-freedom to the CPhy-ML framework to clarify why the drone exhibits a particular behaviour or follows a particular control strategy. The goals of the CPhy-ML framework are effectively achieved by collecting a rich and heterogeneous dataset that persistently excites the model architecture for good generalisation.

[1]School of Aerospace, Transport and Manufacturing, Cranfield University, MK43 0AL Bedford, UK. ✉e-mail: adolfo.perrusquia-guzman@cranfield.ac.uk

The proposed CPhy-ML framework achieves state-of-the-art performance whilst including additional information of the flight physics. A sequential algorithmic tools are provided to predict drone's intention in accordance with the machine learning task. It is demonstrated how intention can be analysed from observational data and enhanced by adding physics informed models and control information. Therefore, this framework provides a deeper insight into the complex nature of intention and a firm step towards to its smooth integration in current counter drone technologies.

## Results

### Proposed method

To better understand the principles of the proposed CPhy-ML framework, it is first described and narrow the set-up of the drone intention prediction problem. The proposed model deals with two different but complementary definitions of intention: trajectory intention and reward function intention. Trajectory intention is associated to the purpose of use of the drone and the potential trajectory profile that the drone will follow in future time steps. The reward function intention describes the hidden motivation used for the control design; this scalar function is the one that the user wants to optimize in an infinite horizon to accomplish any desired task. In this research, open-access Datasets are used to generate a large amount of synthetic data to train

the proposed models (see Methods: Synthetic Data Generation). Four trajectory intention classes are used throughout the research based on the available open-access datasets, which cover: mapping, point-to-point, package delivery, and perimeter flights. These intention classes are described as follows: (i) mapping flights represent flights where a particular region of interest is mapped from images to form a large top-down representation of the region, (ii) point-to-point flights cover long-term transit flights following a straight line between two distant waypoints, (iii) package delivery represents flights from real-world package delivery flight experiments, and (iv) perimeter flights include flights that the starting and ending location point is the same, that is, they followed a closed-loop perimeter pattern. In addition, synthetic data obtained from simulations (Airsim software and Matlab) and real-world data from personal-use drone are used to model other mission profiles that are not labelled in accordance with the proposed trajectory intention classes (see Supplementary Note 2). In addition, two reward function classes are used: normal and anomalous trajectories. These reward intention classes are determined based on the inferred reward function, which weights each mixture of state and control input trajectories to achieve a desired behaviour.

The overview of the CPhy-ML framework for drone intention inference is depicted in Fig. 1. First, in Fig. 1a, the high complex nature of

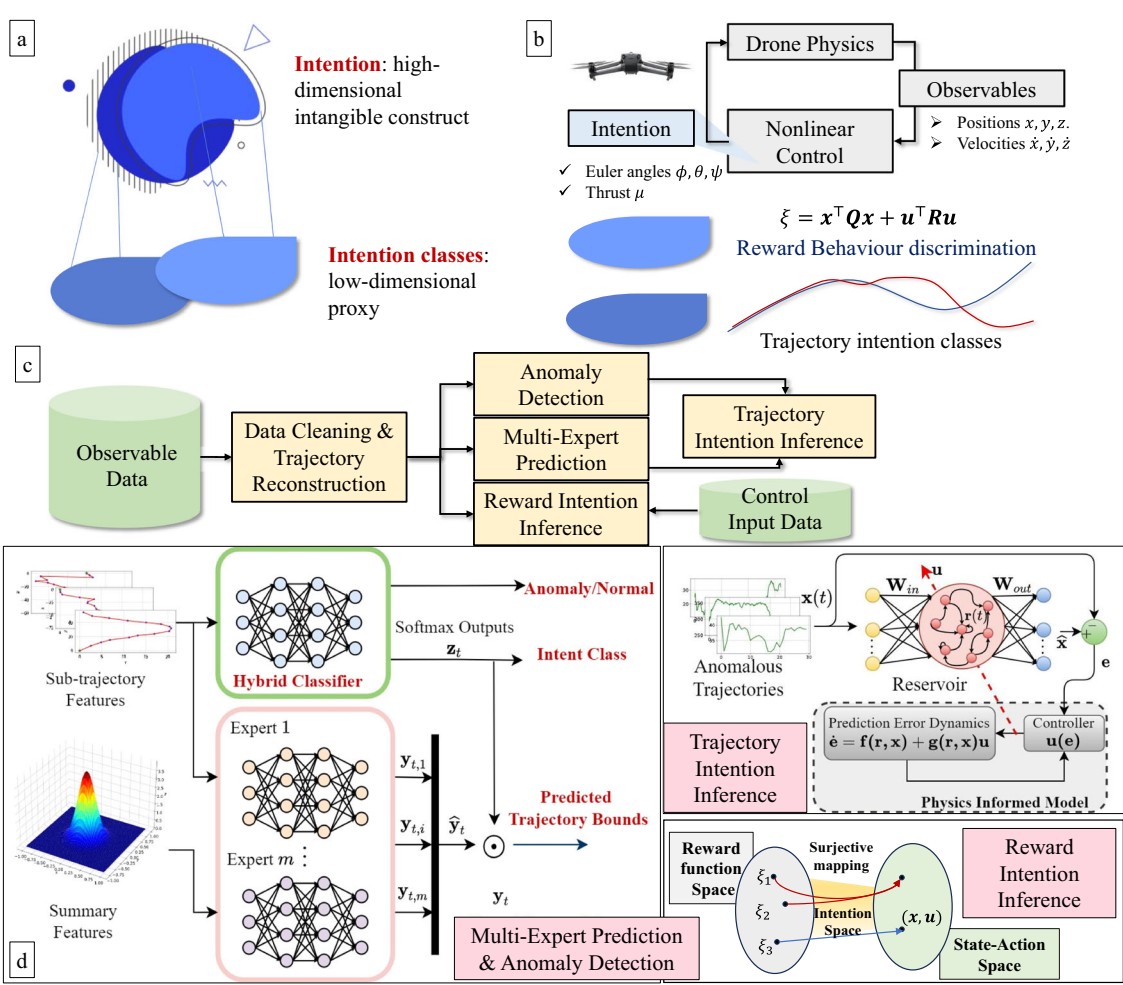

**Fig. 1 | Overview of the control-physics informed machine learning(CPhy-ML) for Drone Intention Inference. a** Illustration of the complex nature of intention and proxy intention classes. **b** High-level view of drone's closed-loop control. Intention is hidden within the control strategy and it is inferred from trajectory observables and reward function design. **c** Flow-diagram of the CPhy-ML intention inference architecture. **d** Description of the main inference and prediction blocks. Multi-Expert Prediction & Anomaly Detection: composed by (1) a Hybrid classifier for intention class prediction and anomaly detection, and (2) *m* autoencoder models for

trajectory reconstruction. The weighted sum between them gives the trajectory bounds of the future airspace that the drone will occupy. Trajectory Intention Inference: Composed of two components: (1) a reservoir computing network for trajectory prediction and (2) a physics informed model for robustness and stability enhancement. Reward Intention Inference: Given by a surjective mapping from the reward function space to the drone's state-action space. The reward function is inferred using an off-policy model-based reward-shaping inverse reinforcement learning architecture.

intention is described as an intangible an unobservable attribute. To give a better understanding of intention, proxy intention classes are defined, which provides a low-dimensional description of the observed performance of the drone. In Fig. 1b, the drone control process is given by the interaction of the drone dynamics and a control architecture. Here, intention is hidden within the control strategy and cannot be measured. However, our proxy definitions of intention classes pave the way to predict intention in a low-dimensional representation. To this end, observables of the drone's states and control data are used to classify intention in accordance with the trajectory profile and the inference of the reward function associated to the control input. Fig. 1c–d give an overview and description of the flow-chart of the proposed framework. First, and to be as realistic as possible, the observable data is constructed from telemetry data[21] of open-access Datasets and real-world data obtained from personal-use drone to generate synthetic radar data trajectories. These trajectories are converted into Sub-trajectory Features and Summary Features of different time windows to ensure real-time detection. Second, a hybrid classifier is used composed by: (1) convolutional bidirectional-LSTM with attention (CBLSTMA) network to classify the trajectory intention class of the input trajectories, and (2) a deep LSTM autoencoder network for trajectory reconstruction and novelty detection. In addition, a deep mixture of experts (DMoE) network is used to predict the bounding boxes of the future trajectory associated to the trajectory intention classes. Here, the output of each expert is weighted by the output probabilities of the neural classifier for the bounding boxes estimation. Each expert is based on a multi-input convolutional neural network. This creates a linear combination between the outputs of the hybrid classifier with the outputs of each expert to correctly predict the future airspace that the drone will occupy. Third, trajectories that are identified as anomalous require individual analysis to predict the future trajectory. This is done by applying reservoir computing methods to obtain high-dimensional features of individual trajectories with less computational effort and enhanced by the incorporation of a physics informed model. Finally, simulated and real-word data obtained from personal use drone are used in a controlled environment to model radio-frequency (RF) data that contains control information. In this scenario, intention is modelled as a surjective mapping from the reward function space (associated to the decision-making controller) to the state-action space of the trajectory data. This reward function is hidden and specifies the task and the desired performance that is injected into the drone. Here, an off-policy model-based reward-shaping inverse reinforcement learning architecture is introduced to uncover the exact reward function based on the sampled trajectories and a linear drone model.

### Novelty detection improves intention classification

It is first presented the results of predicting the drone's trajectory intention class using the Sub-trajectory Features. The results in Fig. 2 show that the hybrid classifier is able to correctly classify similar trajectories within the four possible trajectory intention classes (additional results are reported in Supplementary Figs. 5 and 6). For these trajectories the mean squared error of the reconstruction tends to be small since the testing trajectories exhibit similar patterns or behaviours to the training data (see results of Fig. 2a–b). On the other hand, unseen trajectories tend to have uniform predictions within the classes, which is coherent with the confidence of the trained classifier, i.e., input data that are completely different to the training data reduces the confidence of the classification model (Fig. 2c). In this scenario, the deep autoencoder network cannot reconstruct the trajectory and therefore, the mean squared error of the reconstruction will be large.

Table 1 exhibits the comparison results using conventional classification metrics (Accuracy, Precision, Recall and F1-score) and the mean squared error of the reconstructions between the proposed hybrid classifier and competitive classifiers proposed in the literature (additional results are reported in Supplementary Tables 4 and 5). It is observed that Random Forest classifier shows poor performance across all metrics. In contrast, models with attention such as CBLSTMA and CNNA outperform the classifiers without attention with an approximately 94.67% and 94.37% of

accuracy in the validation set, and 97.95% and 97.56% in the testing set, respectively. One interesting conclusion is that the proposed hybrid classifier shows a similar performance compared with the CBLSTMA classifier with a 94.51% of accuracy in the validation set and 97.75% in the testing set. A slightly degradation across all metrics is observed in the hybrid classifier due to the incorporation of the novelty detector. This compromise is acceptable since it gives to the intention classifier an additional degree of freedom to detect potential malicious drones. The final hyperparameters used in each of the experiments are given in Supplementary Table 3.

It is further evaluated the training and prediction times of each classifier to determine their reliability for real-time classification. Table 2 summarizes the training and prediction times of each neural classifier across all time windows. The results show that the best model CBLSTMA requires 242.91 seconds for training which is almost five times the time of the CNNA with 56.99 seconds. However, the prediction time is practically uniform amongst the other classifiers based on recurrent neural networks with an approximated time of 3.35e−5 seconds. Additional results are given in Supplementary Note 3.

### Multiple experts increases the prediction results

In this experiment, our objective is to predict the future airspace that the drone will occupy in $k$ time steps.

Table 3 shows the performance comparisons between different state-of-the-art regression models averaged across all time windows (extended results are reported in Supplementary Table 7). The results show that the DMoE, where each expert is a multi-input CNN, outperforms the other regression models across all metrics with a $R^2$ coefficient of 0.5105 and 0.7482 for the validation and testing sets, respectively. This result tells us that the DMoE is able to capture more information about the trajectories variability. The results of the single Multi-Input CNN exhibit a $R^2$ coefficient of 0.4290 and 0.3206 for the validation and testing sets, respectively. This allows to conclude that one regression model is not able to capture the richness of different mission profiles. On the other hand, independent regression models for each trajectory intention class can notably improve the regression results but it increases the training time as shown in Table 4. Here, the results show that the training time of the DMoE is less than the multi-input CBLSTMA despite of training four independent multi-input CNN models. The prediction time increases respect to the other models, which can be acceptable due to the regression results improvement. Worth noting that as the number of trajectory intention classes increases, then more experts are required and consequently the prediction time increases. This can be solved by distributing the computational resources or improving the generalization of each expert model for two or more trajectory intention classes. The final hyperparameters used in each of the experiments are given in Supplementary Table 6.

Each output of the DMoE are weighted by the softmax output probabilities of the hybrid classifier to predict the bounding boxes associated to a specific trajectory intention class. Figure 3a–d shows the predicted bounding boxes for each trajectory intention class in a future time-window of 30 s (additional results are reported in Supplementary Note 4). From the overall results of Fig. 3, it is observed that the predicted bounding boxes cover most of the future airspace occupied by the drone. However, as it is expected, the weighted DMoE cannot cover all the variability of each trajectory intention class since it was obtained a $R^2$ coefficient of 0.7482. Here, the challenge is to design an adequate expert model that can capture almost all the variability of the trajectory intention class, or equivalently a high $R^2$ coefficient, in order to obtain high accurate bounding boxes.

### Physics Informed models for prediction stabilization

In this experiment, it is aimed to predict the trajectory that the drone will follow in future time steps. A personal drone is used to conduct real-world testing. The drone comprises a beagle-bone-blue (BBB) chip as its central processor, T-1045 frames and propellers, KV-8816 motors with compatible electronic speed controllers, and a 4-cell 14.8V-5000 mhA LiPo battery. The VICON camera system, composed of 25 well-distributed cameras with

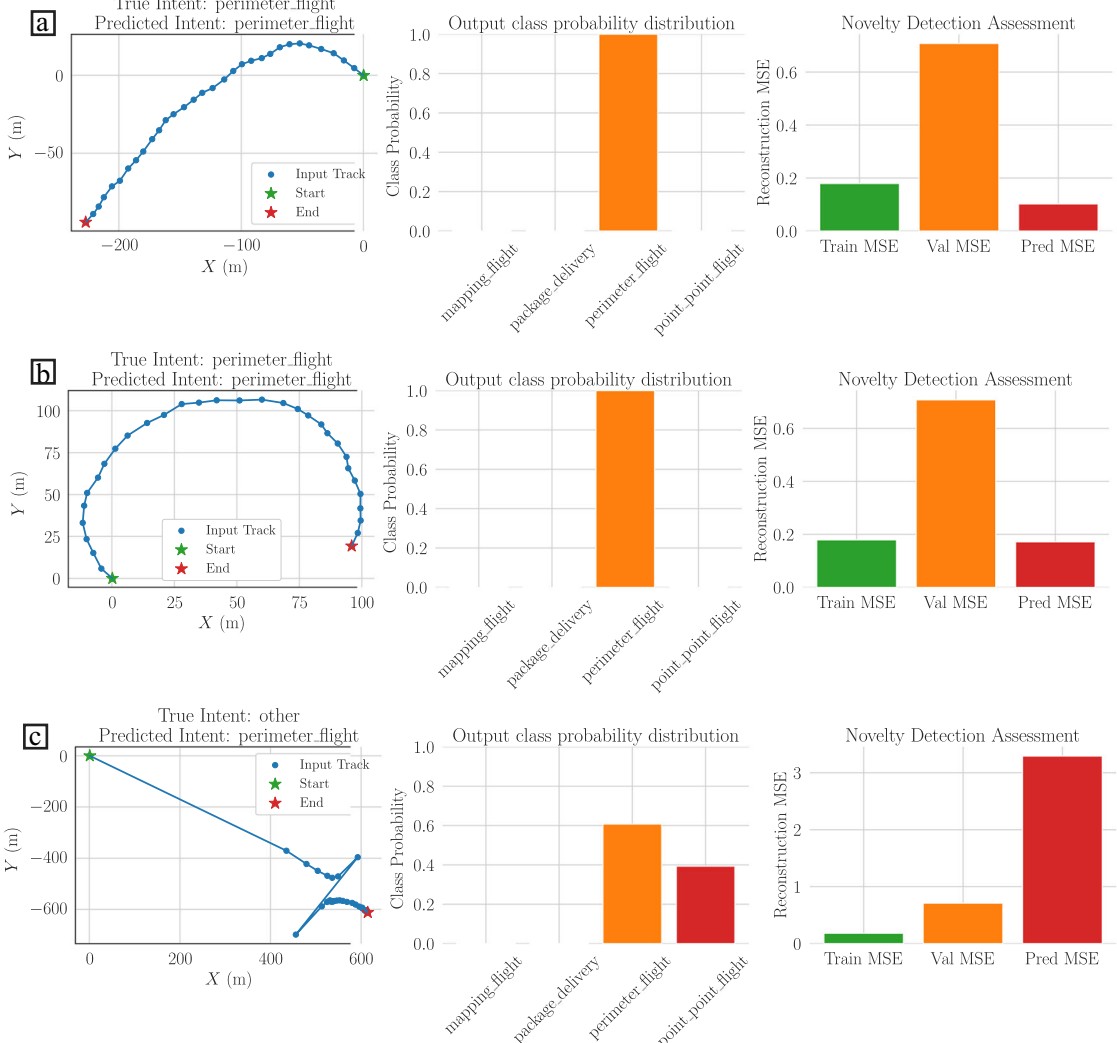

**Fig. 2 | Trajectory Intention Classification example results.** Trajectories are plotted with a blue dotted line, the start point with a green star and the end point with a red star. Each panel is given by one row with 3 plots. **a, b** Shows the results of perimeter flight trajectories. **c** Shows the results of an unknown trajectory profile. The first and second rows of each panel show the classification and novelty detection assessment under the testing dataset using bar plots. The third row of each panel shows the classification and novelty detection assessment under unseen trajectories. The mean squared reconstruction error gives an indicator of potential malicious behaviour.

**Table 1 | Trajectory Intention classification results**

| Model | Validation | | | | | Test | | | | |
|---|---|---|---|---|---|---|---|---|---|---|
| | Accuracy | Precision | Recall | F1-score | Recon MSE | Accuracy | Precision | Recall | F1-score | Recon MSE |
| Random Forest [50] | 0.8756 | 0.8033 | 0.7492 | 0.7590 | – | 0.9205 | 0.9261 | 0.8842 | 0.8933 | – |
| LSTM [51] | 0.9105 | 0.8623 | 0.8611 | 0.8599 | – | 0.9591 | 0.9550 | 0.9630 | 0.9588 | – |
| GRU [52] | 0.9299 | 0.9041 | 0.8868 | 0.8938 | – | 0.9554 | 0.9567 | 0.9380 | 0.9417 | – |
| CBLSTM [53] | 0.9420 | 0.9110 | 0.9099 | 0.9101 | – | 0.9775 | 0.9737 | 0.9832 | 0.9782 | – |
| CBLSTMA [54] | **0.9467** | **0.9205** | **0.9230** | **0.9213** | – | **0.9795** | **0.9761** | **0.9844** | **0.9801** | – |
| CNN [55] | 0.9369 | 0.9009 | 0.9127 | 0.9063 | – | 0.9690 | 0.9633 | 0.9752 | 0.9689 | – |
| CNNA [56] | 0.9437 | 0.9137 | 0.9148 | 0.9140 | – | 0.9756 | 0.9714 | 0.9805 | 0.9757 | – |
| Hybrid Classifier & Novelty Detector (ours) | 0.9451 | 0.9201 | 0.9175 | 0.9179 | 1.4388 | 0.9775 | 0.9735 | 0.9822 | 0.9777 | 0.4511 |

Results are averaged across all time-windows. Best results are in bold.

different resolutions, is used to track the position of the drone. The VICON measurements and transmitting frequency is 120 Hz with a localization error of 0.01–0.5 m.

Table 5 summarizes the mean-squared error of the predictions in different future-time windows. The results show that RC methods with nonlinear readout/decoder model, such as SVM or MLP, perform poorly in the testing data (for predictions 1–100 steps) with a MSE average of 3.1850 and 3.3690, respectively. This result demonstrates that the generalization capabilities of the RC methods lies on the adequate initialization of the reservoir module. On the other hand, an improvement of the RC with linear readout at the testing data is observed across different future time windows by incorporating the physics informed feedback loop. In this scenario, the MSE average is improved from 4.5212 to 1.6471. Here, the physics informed loop constraints the learning manifold and increases the prediction capabilities of classical RC methods. The results for the 1000 steps prediction are tricky. In this case, linear RC tends to diverge for long predictions (which is expected due to the linear nature of the decoder). Here, RC with SVM and MLP encoders exhibit a better MSE results because the prediction is oscillating within a bounded interval such that the MSE is reduced in comparison to the linear RC. However, the predictions of the RC with SVM and MLP decoders are still poor. In the case of the PIRC, an improvement is clearly observed in comparison with the linear RC, i.e., the MSE is improved from 17.3744 to 5.9048. Nevertheless, the predictions are not so accurate as the previous time-windows. The hyperparameters used in the experiment are given in Supplementary Table 8.

Figure 4 shows the results of the trajectory intention prediction algorithm using a noisy real-world trajectory. It is observed, first, that the predicted trajectory is noise-free which is highly appreciated for control purposes. For short future time window predictions (Fig. 4a, b), the RC models show good stable performance. On the other hand, for large future time window predictions (Fig. 4c, d), the linear RC tends to diverge because its region of confidence is reduced. On the other hand, the PIRC enhances the confidence and robustness of the model for larger future time windows (additional results are reported in Supplementary Note 5.4).

## Linear Drone's model: richness is all you need

In this experiment, the control input data is included into the observational data in order to give additional information about the drone's mission profile. Here, the hypothesis consists that drone's intention is embedded within the decision making controller, whose design ensures the drone to have a specific performance or to develop a desired task[22]. This performance is guaranteed by minimizing or maximizing a hidden objective function or reward function which serves as a proxy indicator of a potential misbehaviour.

To this end, first, a dynamic mode decomposition with control (DMDc)[23] model is applied to the RF data to obtain a linear model that preserves the dynamic modes of the real non-linear drone dynamics. It is used the Euler angles: roll $\phi$, pitch $\theta$, and yaw $\psi$; and the total thrust force $\mu$ as control inputs. Using these measurements as control inputs allows to construct a simple discrete linear model. It is observed that the richness of the trajectory is crucial to ensure a good generalization of the model, e.g., point-to-point trajectories are not useful since the dynamic modes of the drone are not excited[24]. In addition, due to the high non-linear dynamics of the drone, then different linear matrices are obtained for different trajectories despite of being from the same drone. To alleviate this problem, the trajectories that exhibit more richness are used to generate the linear model (see Supplementary Note 6.1).

Figure 5 shows the estimated trajectories of the drone's data under the DMDc linear model in closed-loop with an user-design LQR controller. One of the main advantages of this approach is that the nonlinear physics of the drone is transformed into a linear system. This transformation facilitates the prediction analysis with noise suppression. Here, the closed-loop system between the DMDc linear model and the LQR controller is able to track different trajectories accurately and satisfies the small angle condition for the Euler angles control input. Table 6 summarizes the MSE results across all the telemetry data trajectories obtained from custom flights (see RF Sensor). The results show the estimated linear system under the LQR control is capable to estimate accurately both periodic and non-periodic trajectories. Moreover, the inferred states are noise-free which is a requirement in most machine learning techniques to avoid biased predictions. Here, the proposed DMD-LQR approach can be regarded as an effective tool for noise-suppression. Additional results are given in Supplementary Note 6.1.

This approach has two main challenges: (1) the lack of richness in the data which can hinders the acquisition of an accurate linear model, and (2) the control design is sensitive to the model and may require a fine expert tuning. One interesting conclusion is that a more general linear model can be used for prediction purposes by ensuring small angle approximation. This statement is exploited to uncover the hidden reward function.

## Reward function: the most succinct and robust definition of the task

Figure 1 shows that the reward intention is modelled as a surjective mapping between the reward function space to the state-action space. This mapping means that there are different reward functions that can produce the same behaviour in the drone.

In this experiment, it is aimed to uncover the exact hidden reward function associated to the drone's controller using a model-based reward-shaping inverse reinforcement learning (IRL) architecture (see Supplementary Note 6.4). The reward function is modelled as a quadratic function in the states and control weighted by unknown positive semi-definite and

## Table 2 | Training and prediction times of the deep neural classifiers

| Model | Training time (s) | Mean prediction time (s) |
|---|---|---|
| LSTM | 191.34 | 3.63e−5 |
| GRU | 103.1 | 3.38e−5 |
| CBLSTM | 111.29 | 3.35e−5 |
| CBLSTMA | 242.91 | 3.36e−5 |
| CNN | 78.89 | **1.69e−5** |
| CNNA | **56.99** | 1.80e−5 |

Results are averaged across all time-windows. Best results are in bold.

## Table 3 | Trajectory Intention Regression results

| Model | Validation | | | Test | | |
|---|---|---|---|---|---|---|
| | RMSE | MAE | $R^2$ | RMSE | MAE | $R^2$ |
| Multiple Linear Regressor[57] | 143.4732 | 75.3775 | −0.4937 | 137.3331 | 79.542 | −0.7422 |
| Multi-Input BLSTM[58] | 99.2295 | 39.8334 | 0.2946 | 95.6056 | 43.2395 | 0.4576 |
| Multi-Input CNN[59] | 89.6387 | 37.3139 | 0.4290 | 85.3510 | 40.3586 | 0.3206 |
| Multi-Input CBLSTMA[60] | 96.5572 | 38.9600 | 0.3477 | 91.5615 | 42.1395 | 0.1829 |
| DMoE (ours) | **84.1150** | **27.2587** | **0.5105** | **70.2524** | **28.0174** | **0.7482** |

Results are averaged across all time-windows. Best results are in bold.

definite weight matrices $Q$ and $R$ such that the controller is given by a continuous-time LQR. The performance of the proposed reward shaping IRL is compared against the gradient IRL[25] and model-based IRL[26] under diagonal and non-diagonal weight matrices. Knowledge of the exact weight matrix $R$ is assumed for both the gradient and model-based IRL algorithms, with an initial weight matrix $Q_0 = \mathbf{0}_n$. The gradient IRL uses a learning rate of $\gamma = 0.1$. For the reward-shaping IRL, random initial weight matrices are proposed.

Table 7 summarizes the root mean squared spectral norm error (RMSSNE) results of the IRL algorithms under diagonal and non-diagonal weight matrices after 5,000 episodes. Despite the weight matrix $R$ is assumed known for both the gradient and model-based IRL approaches, the results show they cannot converge to the real values. Furthermore, the initial weight matrix $Q_0$ plays a major role in the convergence of the respective IRL algorithm. On the other hand, the proposed reward-shaping IRL architecture overcomes these issues and simultaneously estimate both $Q$ and $R$ and verify the convergence to their near real values.

Figure 6 shows the convergence results of the proposed IRL architecture using the spectral norm error of the control gain, kernel matrix, and weight matrices. Stable and fast convergence results are reported for different reward function structures (Fig. 6a for diagonal weight matrices and Fig. 6b for non-diagonal weight matrices). In addition, each element of the reward function weight matrices converge approximately to their real values in the limit. Here, the results are consistent with the observed behaviours whose values can be reduced by increasing the number of episodes or setting the initial weight matrices close to the real values. A notable decrease is

observed in the RMSSNE from 2.391 and 1.9802 to 0.1942 for diagonal weight matrices and from 6.2848 and 3.3747 to 0.3229.

Table 8 shows that the reward function gives an indicator of misbehaviour given different mission profiles. Anomalous trajectories are recognized in two main scenarios: (1) large tracking error due to disturbances or fast trajectories (Fig. 7b), and (2) control inputs that violates the small angle condition which are infeasible under a linear drone model. One interesting conclusion is that dissipative forces such as drag forces, can attenuate the effect of the weight matrices under high velocity profiles. Conversely, the mean values of the reward function under low velocity profiles are degraded in presence of dissipative terms. The controller plays a fundamental role in the disturbance attenuation process and consequently helps to reduce the suspicion of anomalies (Fig. 7a).

## Discussion

A CPhy-ML framework is introduced to uncover two definitions of intention associated to their intended trajectory and the control objective associated to a reward function. The framework possesses strong inference capabilities with competitive, robust and stable results. The models within the CPhy-ML framework achieve this by combining the capabilities of data-driven methods with physics informed models for learning manifold regularisation. A discussion of the elements of the proposed CPhy-ML framework are given in Methods.

For trajectory intention classification tasks, the CPhy-ML framework can bring great value. This is because it can capture high-dimensional patterns to correctly classify the intention class with competitive results, such as CBLSTMA, while simultaneously recognizing unknown mission profiles. The framework uses a hybrid-classifier which uses the CBLSTMA as classifier with a novelty detector based on a LSTM autoencoder. Here, the hybrid classifier is flexible in the sense that different classifiers can be incorporated into the network, whilst the novelty detector is modified accordingly to the encoder layers of the classifier. A discussion of the hybrid classifier is given in Methods. On the other hand, trajectory intention regression is highly benefited by the CPhy-ML framework by incorporating a set of expert regression models for each trajectory intention class. The use of the bounding boxes gives a visual notion of the future airspace that the drone will occupy using the outputs of the hybrid classifier and the regression experts. Moreover, for trajectory intention prediction, reservoir computing methods used in this framework are a suitable choice to model

**Table 4 | Training and prediction times of the regression models**

| Model | Training time (s) | Mean prediction time (s) |
|---|---|---|
| Multi-Input BLSTM | **148.31** | 2.31e−5 |
| Multi-Input CNN | 171.37 | **1.48e−5** |
| Multi-Input CBLSTMA | 1092.22 | 3.10e−5 |
| DMoE (ours) | 655.25 | 6.19e−5 |

Results are averaged across all time-windows. Best results in bold.

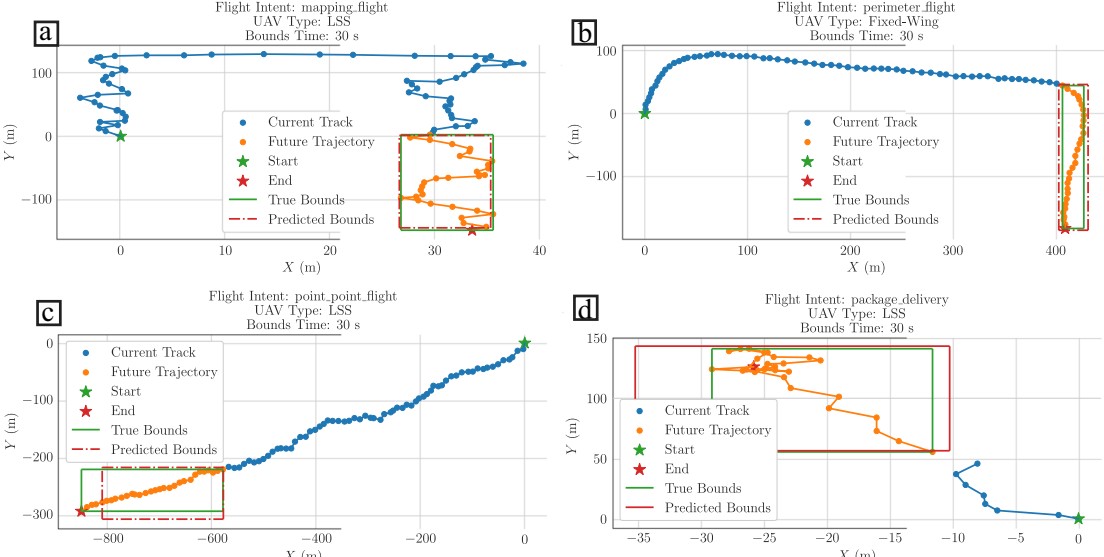

**Fig. 3 | Trajectory Intention Regression example results.** The blue dotted line is the current trajectory track, the orange dotted line is the future trajectory, the start point is given by a green star, and the end point is given by a red star. The green solid line bounding box corresponds to the true future airspace and the red dotted line bounding box corresponds to the predicted future airspace that will occupy the drone. **a** Results of a mapping flight. **b** Results of a perimeter flight. **c** Results of a point-to-point flight. **d** Results of a package delivery flight.

**Table 5 | Trajectory intention prediction results**

| Prediction window (steps) | Mean squared error (MSE) | | | | | | | |
|---|---|---|---|---|---|---|---|---|
| | RC Linear[61] | | RC SVM[62] | | RC MLP[63] | | PIRC (ours) | |
| | Train | Test | Train | Test | Train | Test | Train | Test |
| 1 | 0.0622 | 0.1978 | **0.0584** | 3.1333 | 0.1528 | 1.3930 | 0.0817 | <u>**0.1973**</u> |
| 10 | 0.0629 | <u>**0.1475**</u> | **0.0587** | 3.1334 | 0.2186 | 1.1086 | 0.0914 | 0.1481 |
| 100 | 0.0647 | 0.3653 | **0.0585** | 3.1308 | 0.1298 | 3.9171 | 0.1358 | <u>**0.3384**</u> |
| 1,000 | 0.0629 | 17.3744 | **0.0579** | <u>**3.3427**</u> | 0.1870 | 7.0575 | 2.1259 | 5.9048 |
| Average | 0.0632 | 4.5212 | **0.0583** | 3.1850 | 0.1720 | 3.3690 | 0.6087 | <u>**1.6471**</u> |

Mean Squared error results across different future time windows and different mission profiles. Best results for training are in bold and best results for testing are in bold and underlined.

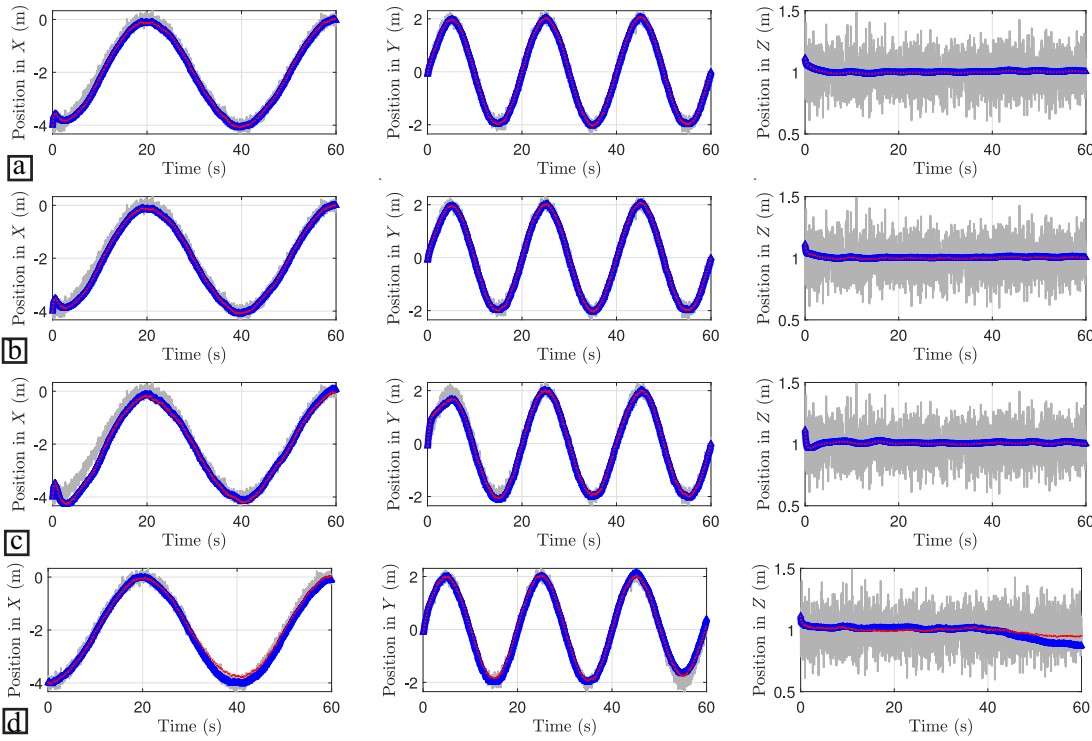

**Fig. 4 | Reservoir computing (RC) results of a single mission profile.** Ground truth data is represented by a solid grey line, the linear RC results are depicted with a blue dotted line with triangle markers, and the red dotted line represents the physics-informed RC (PIRC) results. Each panel is given by one row with 3 plots, and each column defines the predictions of the positions in X, Y, and Z axes, respectively. **a** Prediction results for a future time window of 1 step. **b** Prediction results for a future time window of 10 steps. **c** Prediction results for a future time window of 100 steps. **d** Prediction results for a future time window of 1000 steps.

the high variability of the drone dynamics as a continuous-time differential equation. This is because RC models enforce the high-dimensional representation capabilities of recurrent neural networks with less computational effort, and provides an elegant and natural mechanism to incorporate physics informed models. The interplay between the representation power of RC methods with physics informed models increases the robustness and prediction precision. A discussion of the trajectory intention predictor based on RC network is given in "Methods". Lastly, reward function inference has been focus of attention from several research communities to incorporate explanations of drone's autonomous decision making. Two reward intention classes are defined based on the reward function values obtained from a particular mission profile. It is shown that the design of the reward function defines a desired behaviour that it is injected to the drone. Therefore, mission profiles that do not meet the behaviour requirements will exhibit highest reward function values which can be attributed to a potential mis-behaviour. Here, the CPhy-ML framework gives an insight in how sources

of explanations can be provided by uncovering the hidden reward function using the proposed off-policy model-based reward-shaping IRL algorithm. A discussion of the model-based reward-shaping IRL is given in Methods. However, more research is required in this area to ensure robust, stable and trustworthy explanations of drone behaviour.

**What are the limitations of the proposed CPhy-ML framework?**
The CPhy-ML framework is limited by the amount of data and its variability (richness). This limitation hinders the accurate generalisation of both the trajectory intention classifier and regression models. Specifically, the low variability of the data can cause a wrong location of the bounding boxes. To prevent this, it is required to collect data associated to other trajectory intention classes and construct an intention dictionary. Moreover, in this research the synthetic data generation was limited to few variations which can be further improved to increase the richness of the data. The prediction time of the DMoE is increased as more trajectory intention classes are

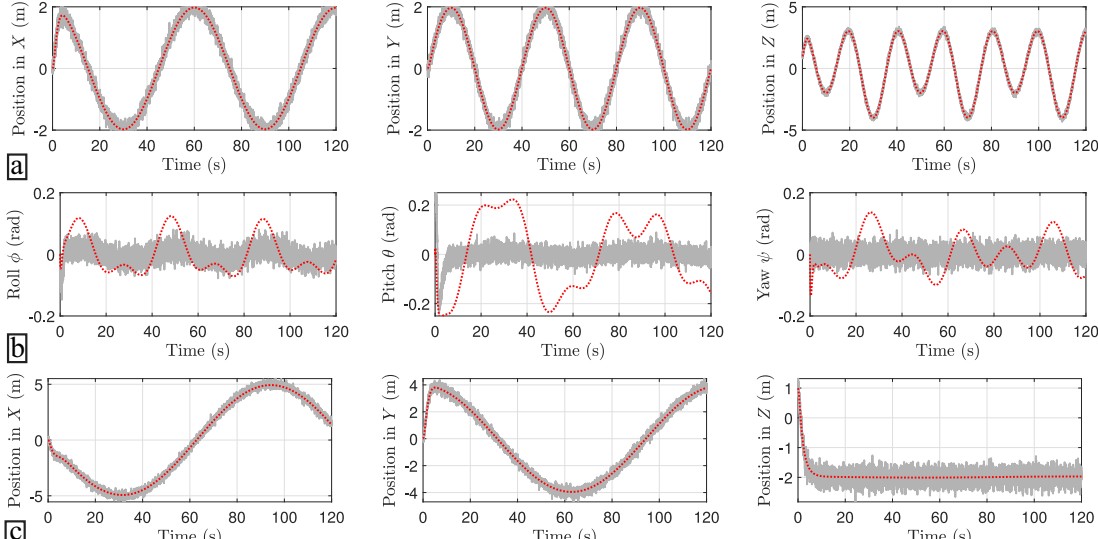

**Fig. 5 | Dynamic mode decomposition (DMD) linear model with linear quadratic regulator (LQR) estimation.** The results with solid gray line corresponds to the raw measurements, while the results in red dotted line stand for the DMD-LQR estimation. Each panel is given by one row with 3 plots. **a** Estimated trajectories of the trained linear model. **b** Euler angles control inputs obtained from the LQR design. **c** Generalization capabilities of the estimated linear model under different trajectories.

**Table 6 | Mean Squared Error (MSE) of the DMD-LQR algorithm across diverse periodic and non-periodic trajectories**

| MSE | | |
|---|---|---|
| **Position axes** | **Periodic trajectories** | **Non-periodic trajectories** |
| X | 0.06198 | 0.05709 |
| Y | 0.06277 | 0.05829 |
| Z | 0.06421 | 0.05947 |
| Average | 0.06299 | 0.05828 |

presented to the network. This can be attenuated by designing experts associated to tasks that pose similar mission profiles.

The CPhy-ML framework offers good trajectory intention prediction results when the RC model is trained under a rich enough data, otherwise its generalisation capability is degraded. This problem can be alleviated by: (1) ensure the data is rich enough to exploit the high-dimensional representation power of RC methods, or (2) design a mixture of RC networks under different reservoir weights to increase the high-dimensional representation heterogeneity and improve the prediction generalisation.

The incorporation of the control input for prediction purposes or reward function inference gives insights of the mission profile. However, its scope is limited for short-term predictions or constant references. In addition, high level of noise can compromise the policy prediction and thus, the inferred reward function. This problem requires additional analysis to first attenuate the noise and second to increase the prediction capabilities using the control information. This can be solved by incorporating state estimation and parameter identification techniques such as closed-loop output error techniques[27] and new methodologies for inverse reinforcement learning based on model predictive control and experience inference[28].

One additional limitation or area of opportunity consists in the incorporation of the values of the reward function to constrain the learning manifold. On the one hand, this measurement is not a common data provided by on-board sensors which can be view as a limitation. On the other hand, this additional data can serve to reform new drone regulations procedures in function of the states and control input information.

## Methods

The models and notations used in our experimental results are summarized in Supplementary Note 1.

### Synthetic data generation

Two different sources of data are considered within the scope of this research: non-cooperative radar[29,30] and radio frequency (RF) sensor measurements[31].

**Non-cooperative Radar - Off-line data**. A custom radar simulation process is developed based on the Stone Soup software[32] and the open-access telemetry data discussed at Datasets. These datasets are pre-processed as follows: (1) converting latitude, longitude, and altitude into local Cartesian coordinates; (2) removing unwanted periods (e.g., take-off, on ground); and (3) down/up-sampling to 1 Hz. It is assumed that each measurement of the simulated radar has Gaussian noise.

Data augmentation is applied by changing the noise of the nonlinear measurement process and radar location. Two different process noise intensities of 1.0 and 3.0 and seven different relative radar locations are used to generate a pool of heterogeneous trajectories based on the telemetry data input. An extended Kalman Filter (EKF) is used to obtain the final simulated radar tracks based on the following nonlinear model per axis

$$\boldsymbol{x}_t = \boldsymbol{F}_t \boldsymbol{x}_{t-1} + \boldsymbol{\omega}_t, \ \boldsymbol{\omega}_t \sim \mathcal{N}(\boldsymbol{0}, \boldsymbol{Q}_t)$$

$$\boldsymbol{x}_t = \begin{bmatrix} x_{\text{pos}} \\ x_{\text{vel}} \end{bmatrix}, \boldsymbol{F}_t = \begin{bmatrix} 1 & dt \\ 0 & 1 \end{bmatrix}, \boldsymbol{Q}_t = \begin{bmatrix} \frac{dt^3}{3} & \frac{dt^3}{2} \\ \frac{dt^3}{2} & dt \end{bmatrix} q. \quad (1)$$

where $x_{\text{pos}}$ and $x_{\text{vel}}$ are the Cartesian position and velocity of the x-axis, $q$ is the velocity noise diffusion constant which is set to 0.1 to obtain smooth track estimations that closely matches with the original flight trajectories. Additional information is given in Supplementary Note 2.1.

**RF sensor**. In this scenario, telemetry data obtained from custom flights are used to model real-time tracking obtained from RF sensors[33], e.g., the DJI's Aerospace RF sensor which can detect and track all DJI RF Drones (70%—estimated market share in the drone industry). Here, the

## Table 7 | Objective function inference results

| RMSSNE | | | | | | |
|---|---|---|---|---|---|---|
| $e_i$ | Gradient IRL | | Model-(ased IRL | | Reward (haping IRL (ours) | |
| | Diagonal weight matrices | Non-diagonal weight matrices | Diagonal weight matrices | Non-diagonal weight matrices | Diagonal weight matrices | Non-diagonal weight matrices |
| $\|\boldsymbol{K}_{i+1} - \boldsymbol{K}_i\|$ | 0.0096 | 0.0044 | **0.0072** | **0.0012** | 0.0273 | 0.0328 |
| $\|\boldsymbol{P}_{i+1} - \boldsymbol{P}_i\|$ | **0.0061** | **0.0024** | 0.0136 | 0.0039 | 0.0641 | 0.0740 |
| $\|\boldsymbol{Q}_{i+1} - \boldsymbol{Q}_i\|$ | 0.0562 | 0.0322 | **0.0486** | **0.0179** | 0.0907 | 0.1093 |
| $\|\boldsymbol{R}_{i+1} - \boldsymbol{R}_i\|$ | – | – | – | – | **0.0197** | **0.0424** |
| $\|\boldsymbol{K}_i - \boldsymbol{K}\|$ | 0.0480 | 0.0108 | **0.0153** | **0.0099** | 0.0649 | 0.0724 |
| $\|\boldsymbol{P}_i - \boldsymbol{P}\|$ | 2.3886 | 6.5916 | 1.9761 | 3.3759 | **0.1331** | **0.1633** |
| $\|\boldsymbol{Q}_i - \boldsymbol{Q}\|$ | 2.3910 | 6.2848 | 1.9802 | 3.3747 | **0.1942** | **0.3229** |
| $\|\boldsymbol{R}_i - \boldsymbol{R}\|$ | – | – | – | – | **0.1051** | **0.2693** |

Root mean squared spectral norm error (RMSSNE) results across different weight matrices and IRL algorithms. Best results are in bold.

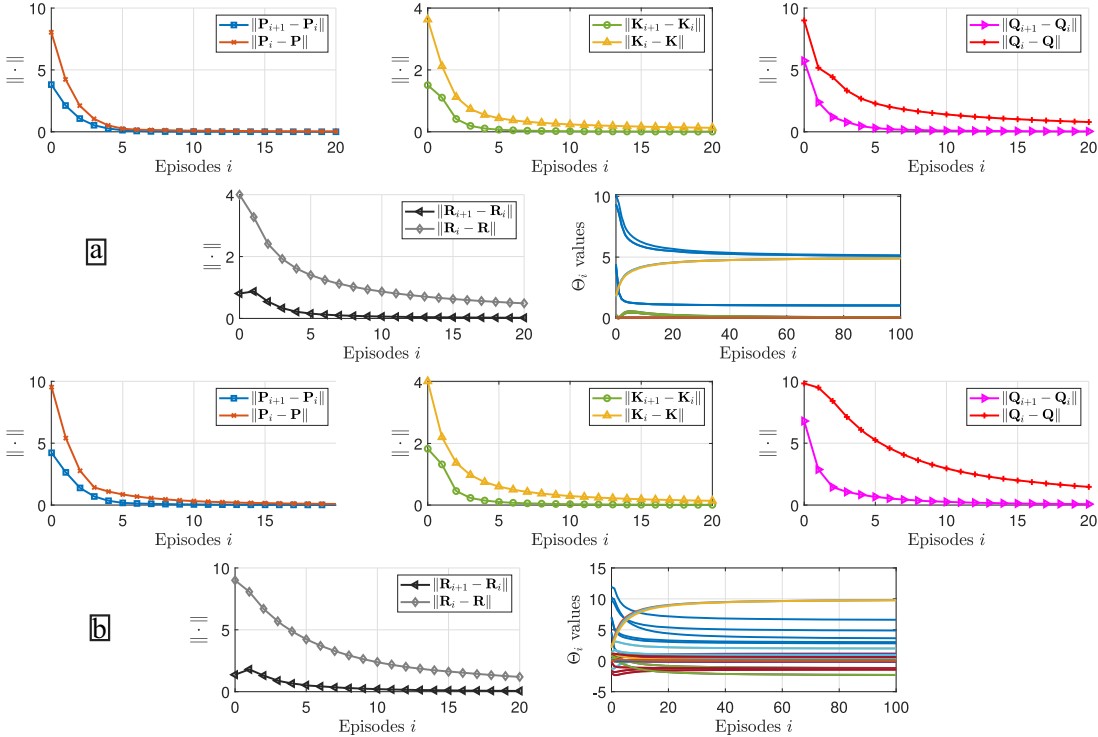

**Fig. 6 | Convergence results of the Reward function Inference Algorithm.** Spectral norm error of the kernel matrix $\boldsymbol{P}_i$ in blue and orange lines with square and cross markers. Control gain $\boldsymbol{K}_i$ in green and yellow lines with circle and triangle markers. Spectral norm error of the reward weight matrix $\boldsymbol{Q}_i$ with pink and red lines with right triangle and plus markers. Spectral norm results of the weight $\boldsymbol{R}_i$ with black and gray lines with left triangle and diamond markers. Each panel is given by two consecutive rows with 5 plots. The fifth figure of each panel shows the convergence of the elements of $\boldsymbol{Q}_{i+1}$ and $\boldsymbol{R}_{i+1}$ in $\boldsymbol{\Theta}_i$ to their approximately exact values $\boldsymbol{Q}$ and $\boldsymbol{R}$, respectively. **a** Results for diagonal weight matrices. **b** Results for non-diagonal weight matrices.

measurements are assumed to belong from a continuous-time model[34] of the form

$$\dot{\boldsymbol{x}} = \begin{bmatrix} \boldsymbol{0}_{3\times3} & \boldsymbol{I}_3 \\ \boldsymbol{0}_{3\times3} & \boldsymbol{0}_{3\times3} \end{bmatrix} \boldsymbol{x} + \begin{bmatrix} \boldsymbol{0}_3 \\ \frac{1}{m}(\boldsymbol{F} + \boldsymbol{F}_g) \end{bmatrix} := \boldsymbol{f}(\boldsymbol{x}, \boldsymbol{u}),$$

$$\boldsymbol{F} = \begin{bmatrix} \mu(\sin\phi\sin\psi + \cos\phi\cos\psi) \\ \mu(-\sin\phi\cos\psi + \cos\phi\sin\theta\sin\psi) \\ \mu\cos\phi\cos\theta \end{bmatrix}, \quad \boldsymbol{F}_g = \begin{bmatrix} 0 \\ 0 \\ -mg \end{bmatrix}. \quad (2)$$

where $\boldsymbol{x} = [x, y, z, \dot{x}, \dot{y}, \dot{z}]^\top$ is the state vector composed of the linear positions and velocities in the Cartesian space, $m$ is the mass of the drone, $g$ is the gravity acceleration, $\mu$ is the total thrust, and $\phi$, $\theta$, and $\psi$ denote the roll, pitch and yaw Euler angles[35]. In this scenario, it is assumed that measurements of the state $\boldsymbol{x}$ and the inputs $\boldsymbol{u} = [\phi, \theta, \psi, \mu]^\top$ are available with some Gaussian distributed noise[36]. Additional information is given in Supplementary Note 2.

### Hybrid intention classifier
**Sub-trajectory features.** The trajectory tracks are processed into several sub-trajectories each with an associated intention label. This is to cover

the needs of real-time prediction using partial information of the complete trajectory and observe the robustness of the deep models. Four different window sizes are used in this research: 8, 16, 32 and 64 s.

The output sub-trajectories are split into training, validation and testing partitions using 75%, 15% and 10% of the total data, respectively. After splitting into suitable partitions, each sub-trajectory is pre-processed as follows: (i) change of coordinates to start at the origin $(x, y, z) = (0, 0, 0)$; (ii) standardisation of numerical features to have approximately zero mean and standard deviation of one; and (iii) one-hot encoding of each categorical variable within each sub-trajectory.

**Intention classifier.** A convolutional bidirectional-LSTM with attention[37] network is used as classifier. This network is divided into encoder and classification layers. The encoder ones are comprised of one-dimensional convolutional layers which are applied to the input sequences followed by bidirectional LSTM recurrent layers[38]. The classification part applies attention layers[39] into each output of the recurrent layers followed by a fully connected network using the softmax activation function.

Regularisation is added to the classifier to avoid overfitting. This is done by adding recurrent and dense dropout between certain layers in the classification network. An early stopper is used in the training phase to monitor the validation loss over time. Here, the training phase is stopped after no improvement is observed in a specified number of epochs. This helps to identify the best model weights before overfitting the training data.

**Novelty detector.** A novelty detector network is used to determine the similarity of novel or unseen flight profiles with respect to the training classes. The novelty detector is based on a deep autoencoder architecture

**Table 8 | Mean values of the reward function**

| Reward function $\xi = (x^d - x)^\top Q(x^d - x) + u^\top R u$ | | |
|---|---|---|
| **Trayectory** | **Mean value without drag force** | **Mean value with drag force** |
| Fixed point | 0.1073 | 0.1077 |
| Helix | 0.3992 | 0.4432 |
| Circular | 2.3756 | 2.2749 |
| Infinity-shape | 0.0879 | 0.1392 |
| Fast Helix | **313.2480** | **290.4161** |
| Fast Circular | **85.9878** | **79.6712** |
| Fast Infinity-shape | **142.0225** | **130.5464** |

Results using $Q = \mathrm{diag}\{1, 1, 1, 5, 5, 5\}$ and $R = 5I_3$. Anomalous trajectories are highlighted in bold.

which uses the encoder layers of the Intention Classifier followed by a LSTM decoder network[40].

**Training loss function for classification and novelty detection.** The hybrid classifier training is performed with a supervised composite loss function composed by a weighted sum between a categorical cross-entropy loss (for trajectory intention classification) and a mean-squared error loss (for anomaly detection based on the reconstruction error). The hybrid loss is given by

$$
\begin{aligned}
\mathcal{L}(\boldsymbol{y}, \widehat{\boldsymbol{y}}) &= -\frac{1}{N} \sum_{n=1}^{N} \sum_{k=1}^{K} y_{nk} \ln(\widehat{y}_{nk}), \\
\mathcal{L}(\boldsymbol{X}, \widehat{\boldsymbol{X}}) &= \frac{1}{NMT} \sum_{n=1}^{N} \sum_{m=1}^{M} \sum_{t=1}^{T} (x_{nmt} - \widehat{x}_{nmt})^2, \\
\mathcal{L}_{\mathrm{hyb}} &= \alpha \mathcal{L}(\boldsymbol{y}, \widehat{\boldsymbol{y}}) + (1 - \alpha) \mathcal{L}(\boldsymbol{X}, \widehat{\boldsymbol{X}})
\end{aligned}
\tag{3}
$$

where $\alpha \in (0, 1)$ defines the weight importance between each loss. In this research $\alpha$ is set to $\alpha = 0.95$ to prioritize the intention classification rather than the input-sequences reconstruction.

**Deep mixture of experts for trajectory intention regression**

**Summary features.** This model has two inputs given by: (1) Sub-trajectory Features used in the hybrid classifier; and (2) Summary Features associated to each particular intention class. These summary features are determined by the mean, standard deviation, minimum and maximum points of each sub-trajectory feature.

**Regression experts.** An assembled architecture based on $m$ multi-input convolutional neural network is used to model the trajectories of each trajectory intention class. Two inputs has this architecture: (i) 1D convolutional layers for the sub-trajectory input sequences; and (ii) Deep Neural Network (DNN) layers for the Summary Features. The embeddings from each of these networks are concatenated and fed into the final fully connected layers for regression.

**Training loss function for regression.** The Huber loss is used as training loss to combine the benefits of both the mean-squared error and mean-absolute error. This is defined using a single scalar output response $y_i$ and a predicted response $\widehat{y}_i$ as

$$
\mathcal{L}_{\mathrm{huber}}(y_i, \widehat{y}_i) = 
\begin{cases}
\frac{1}{2}(y_i - \widehat{y}_i)^2 & \text{if } |y_i - \widehat{y}_i| \leq \delta \\
\delta|y_i - \widehat{y}_i| - \frac{1}{2}\delta^2 & \text{if } |y_i - \widehat{y}_i| > \delta,
\end{cases}
\tag{4}
$$

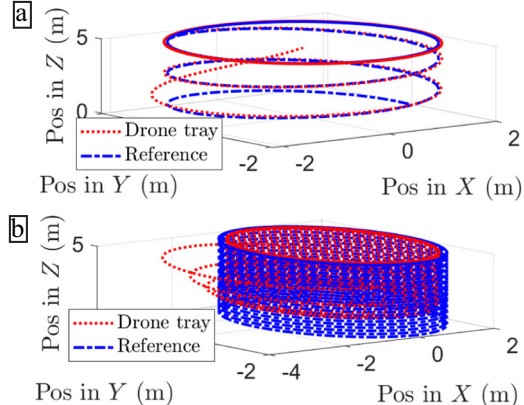
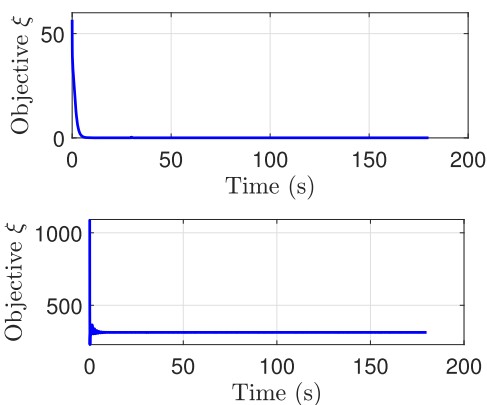

**Fig. 7 | Example of the reward function values under an helix mission profile.** Each panel is given by a row with 2 plots. **a** Slows helix trajectory with red and blue dotted lines and the respective reward function values in solid blue line. **b** Tracking and reward function values for a fast helix trajectory under the same line styles and colours.

where $\delta$ is a threshold that switches between the $L_1$ and $L_2$ errors. The default value of $\delta = 1$ is used during training.

## Trajectory intention prediction

Anomalous trajectories require further analysis that cannot be provided by the deep mixture of experts network. Here, it is more natural to model the measurement process as a continuous differential equation[17]. To this end, each future predicted trajectory $y$ is assumed to be approximated by the following reservoir computing network[41]

$$\begin{aligned}\widehat{y} &= W_{out}r + w \\ \dot{r} &= \sigma(\mathcal{A}r + W_{in}x),\end{aligned} \tag{5}$$

where $W_{in}$ are the input weights, $W_{out}$ are the decoder/readout weights, $w$ is the bias vector, $\mathcal{A}$ denotes the reservoir weights, $x$ are the input trajectories, $r$ are the reservoir states, and $\sigma(\cdot)$ is a S-shaped function. In this research $\sigma(\cdot)$ is set to $\tanh(\cdot)$.

**Weights calculations.** The input weights $W_{in}$ are randomly generated between the interval $W_{in} \in [-1, 1]$. The reservoir weights are computed as $\mathcal{A} = -\frac{1}{2}(A_0^\top A_0)$, for some random generated matrix $A_0$ drawn from a standard normal distribution. The decoder weights $W_{dec} := [W_{out}|w]$ are obtained by the minimization of the following convex optimization problem

$$W_{dec}^* = \underset{\{W_{out}, w\}}{\operatorname{argmin}} \frac{1}{2} \| W_{dec}\bar{R} - Y \|_{\mathcal{F}}^2 + \lambda \| W_{out} \|_{\mathcal{F}}^2, \tag{6}$$

where $\bar{R} = \begin{bmatrix} r_1 & \cdots & r_T \\ 1 & \cdots & 1 \end{bmatrix}$, $Y = [y_1, \cdots, y_T]$ are matrices of the reservoir states $r_i$ and trajectory predictions $y_i$ in $i = 1, \ldots, T$ time steps, respectively. $\lambda$ is a regularisation scalar which is set to $\lambda = 0.5$.

**Reservoir weights enhancement via physics informed model.** The standard reservoir computing method can suffer of poor representation capabilities due to the random initialization of the reservoir weights $\mathcal{A}$ and input weights $W_{in}$. To enhance the performance of the network, a physics informed feedback[42,43] is used to update the reservoir weights. Here, the reservoir computing scheme is modified into

$$\begin{aligned}\widehat{y} &= W_{out}r + w, \\ \dot{r} &= \sigma((\mathcal{A} + \mathcal{B})r + W_{in}x),\end{aligned} \tag{7}$$

where $\mathcal{B}$ are the new weights provided by the physics informed model. This weights are computed by the following set of elements

$$\begin{aligned} f(r, x) &= -W_{out}D_\sigma(z_0)\mathcal{A}\tilde{r}, \\ g(r, x) &= -W_{out}D_\sigma(z_0) \otimes \tilde{r}^\top, \\ u &= -g^\dagger(x, r)(f(x, r) + \mathcal{K}e), \\ B_1 &= \operatorname{mat}(u), \\ \mathcal{B} &= -\frac{1}{2}\left(B_1^\top B_1 + \epsilon I_r\right) \end{aligned} \tag{8}$$

where $D_\sigma(z_0) = \frac{\partial \sigma(z)}{\partial z}|_{z=z_0}$ is the gradient of $\sigma(\cdot)$ with respect to $z$ and evaluated at the vector $z_0 = (\mathcal{A} + \mathcal{B})r + W_{in}x$[44]. The prediction error is defined as $e = \widehat{y} - y$. The reservoir states error is defined by $\tilde{r} = r - W_{out}^\dagger(y - w)$, and $\mathcal{K}$ is a positive definite diagonal matrix whose values are set small enough to prevent noise excitation. The scalar $\epsilon > 0$ is set small enough for short predictions and relatively large for long predictions. In this research, the value of $\epsilon$ is setted o $\epsilon = 0.0001 f_t$ where $f_t$ is the future time window prediction. Algorithm derivation and details are given in Supplementary Note 5).

## Drone's linear modelling

From RF data, the following matrices are constructed

$$X = \begin{bmatrix} | & & | \\ x_1 & \cdots & x_{i-1} \\ | & & | \end{bmatrix}, X' = \begin{bmatrix} | & & | \\ x_2 & \cdots & x_i \\ | & & | \end{bmatrix}, Y = \begin{bmatrix} | & & | \\ u_1 & \cdots & u_{i-1} \\ | & & | \end{bmatrix}.$$

Dynamic mode decomposition[18] with control (DMDc) is applied to estimate a linear representation of the drone dynamics of the form

$$x_{k+1} = A_D x_k + B_D u_k, \tag{9}$$

where $A_D$ and $B_D$ are the discrete linear matrices associated to a specific drone.

**Dynamic mode decomposition with control (DMDc).** The following set of linear equations are constructed

$$\begin{aligned} X' &= A_D X + B_D Y, \\ &= G\Omega, \end{aligned} \tag{10}$$

where $G = [A_D, B_D]$ and $\Omega = [X^\top, Y^\top]^\top$. Singular value decomposition (SVD) is applied in the matrix $\Omega$ to compute the matrix $G$ as

$$G = X'V\Sigma^{-1}U^*, \tag{11}$$

where $U$ and $V$ are the left and right singular matrices, and $\Sigma$ denotes the matrix of singular values. A low-rank approximation is commonly used in most of the applications using DMD due to the high number of states an the presence of noise. Then, the low-rank approximation is given by

$$G \approx X'\widetilde{V}\widetilde{\Sigma}^{-1}\widetilde{U}^*, \tag{12}$$

where $\widetilde{U}, \widetilde{V}, \widetilde{\Sigma}$ are low-rank approximations of $U, V$ and $\Sigma$, respectively. A low-dimensional representation can also be applied to the estimated matrices. However, this model leads to loss of information with an unsatisfactory representation of the drone dynamics. In this research, it is used either (11) or (12) to compute the linear matrices $A_D$ and $B_D$. Therefore, the linear model predictor is given by

$$\widehat{x}_{k+1} = A_D\widehat{x}_k + B_D\widehat{u}_k, \tag{13}$$

where $\widehat{x}_k$ is the state prediction of the model and $\widehat{u}_k$ is an estimated control input.

**Control policy linear modelling.** A discrete linear quadratic regulator (LQR)[45] is used to obtain $\widehat{u}_k$ in (13). This controller will ensure reference tracking and will facilitate the prediction of the future trajectory. The LQR controller fulfils the next discrete algebraic Riccati equation (DARE)[25]

$$A_D^\top P_D A_D + Q_D - A_D^\top P_D B_D(R_D + B_D^\top P_D B_D)^{-1}B_D^\top P_D A_D = P_D, \tag{14}$$

for some symmetric and positive definite matrix $P_D = P_D^\top$ and user-design positive definite and symmetric matrices $Q_D = Q_D^\top$ and $R_D = R_D^\top$. Hence, the final controller is

$$u_k = (R_D + B_D^\top P_D B_D)^{-1}B_D^\top P_D A_D(x_k^f - \widehat{x}_k) = K_D(x_k^f - \widehat{x}_k), \tag{15}$$

where $x_k^f$ denotes the noise-free measurement of $x_k$ and models the desired reference that is following the drone. This signal is obtained from either using the RC computing method or any signal processing technique. Notice that the filtered measurement $x_k^f$ can be used instead of the noisy measurement $x_k$ to construct the matrices $X$ and $X'$. However, the noise

provides excitation to the DMDc model such that the linear matrices outputs have a better representation of the drone's dynamics.

## Reward function inference

Without loss of generality and in view of the results obtained from the DMD method, a simulated continuous-time linear system[46] is constructed to model the drone's dynamics

$$\dot{x} = Ax + Bu$$
$$y = Cx + v, v \sim \mathcal{N}(0, R) \tag{16}$$

with

$$A = \begin{bmatrix} 0_{3\times3} & I_3 \\ 0_{3\times3} & 0_{3\times3} \end{bmatrix}, B = \begin{bmatrix} 0_3 & 0_3 & 0_3 \\ 0 & -g & 0 \\ g & 0 & 0 \\ 0 & 0 & \frac{1}{m} \end{bmatrix}, C = I_n,$$

where $x = [x, y, z, \dot{x}, \dot{y}, \dot{z}]^\top$ denotes the vector of Cartesian positions and velocities, $u = [\phi, \theta, \mu]^\top$ is the control input composed by the Euler angles: pitch and roll; and the total thrust produced by the drone, $g = 9.81\text{ms}^{-2}$ is the gravitational acceleration and $m = 0.467$ kg is the mass of the drone. This linear system is obtained around the hover flight condition[47].

The small angle condition can be achieved by designing a linear quadratic regulator (LQR) control[48] of the form

$$u = K(x^d - x) = R^{-1}B^\top P(x^d - x), \tag{17}$$

that minimizes the infinite horizon cost

$$J = \int_t^\infty \left( (x^d - x)^\top Q(x^d - x) + u^\top Ru \right) d\tau,$$

where $Q = Q^\top$ and $R = R^\top$ are positive definite unknown weight matrices, and $P = P^\top$ is the solution of an algebraic Ricatti equation (ARE). The term $\xi(x^d, x, u) = (x^d - x)^\top Q(x^d - x) + u^\top Ru$ is known as the objective function, utility function or reward function, and is the most succinct, transferable and robust definition of the task that the drone aims to perform[49]. Uncovering this objective function provides causal information of why the drone exhibits a particular behaviour and its final goal intent. To this end, it is collected $\iota$ measurements of the states $X$, the control input $\Upsilon$, the desired reference $X^d$ and the respective reward function values $\Xi$.

**Policy parameterization.** From the collected data, the LQR control gain is estimated following a similar procedure to the DMD method as

$$K_p = \Upsilon V_X \Sigma_X^{-1} U_X^\top, \tag{18}$$

where $V_X$, $U_X$, and $\Sigma_X$ denote the SVD of the matrix $X^d - X$. If the measurements are free of noise then $K_p \equiv K$[25].

**Off-line model-based reward-shaping inverse reinforcement learning.** A model-based reward-shaping inverse reinforcement learning (IRL) approach is proposed to extract the exact hidden reward function. Convergence to the exact weight matrices is achieved by incorporating the feedback of the reward function values that constraints the learning manifold of the IRL architecture. The pseudo-algorithm is summarized in Algorithm 1 (see Supplementary Note 6.4).

**Algorithm 1.** Off-line Model-based reward-shaping inverse reinforcement learning

1: Collect measurements of $X \in \mathbb{R}^{n\times\iota}$, $X^d \in \mathbb{R}^{n\times\iota}$, and $\Xi \in \mathbb{R}^{1\times\iota}$. Select $Q_0 = Q_0^\top > 0$ and $R_0 = R_0^\top > 0$, and a stabilizing gain $K_0$. Set $i = 0$ and a small threshold $\varepsilon_k$.

2: Policy Evaluation. Compute $P_i$

$$0_{n\times n} = (A - BK_i)^\top P_i + P_i(A - BK_i) + K_i^\top R_i K_i \\ - (K_i - K_p)^\top R_i(K_i - K_p) + Q_i. \tag{19}$$

3: Policy Improvement. Compute $K_{i+1}$

$$K_{i+1} = R_i^{-1}B^\top P_i. \tag{20}$$

4: Weights Improvement. Compute $\Theta = [\text{vec}(Q_{i+1})^\top, \text{vec}(R_{i+1})^\top]^\top$

$$\begin{bmatrix} I_{n^2} & -(K_{i+1} \otimes K_{i+1})^\top \\ I_{n^2} & (K_p \otimes K_p)^\top \end{bmatrix} \Theta = \begin{bmatrix} -\text{vec}(A^\top P_i + P_i A) \\ [(X^d - X)^\top \otimes (X^d - X)^\top]^\dagger \Xi^\top \end{bmatrix}. \tag{21}$$

5: Stop the algorithm if $\|K_{i+1} - K_i\| \le \varepsilon_k$, otherwise set $i = i + 1$ and return to Step 2.

The root mean squared spectral norm error (RMSSNE) is used as performance metric to evaluate the inference results.

$$\mathcal{L}_{RMSSNE} = \sqrt{\frac{1}{N}e^\top e} = \sqrt{\frac{1}{N}\sum_{j=1}^N e_j^2}, e_i = \| M_i - N_i \| \tag{22}$$

for any matrices $M_i$ and $N_i$ of appropriate dimension.

## Datasets

**UAV attack dataset.** The UAV attack dataset contains real and simulated flights with a range of different UAV types, including examples of normal flights and system spoofing attacks. The dataset is comprised of 14,295 data samples from 6 different UAVs for waypoints autonomous mapping flights.

**ALFA dataset.** The ALFA dataset provides autonomous flight data conducted with a fixed-wing UAV around a pre-programmed perimeter. The dataset contains 47 autonomous flights classified as normal flights and flights with faults.

**Drone Identification and tracking dataset.** This datasets is provided as part of the International Conference on Military Communication and Information System (ICMCIS). It contains UAV telemetry and radar-based measurements data for a range of fixed-wing and quadcopters flights in long-intervals of flying between point-to-point waypoints.

**Package delivery UAV dataset.** This dataset contains a range of flights of a package delivery UAV. It is comprised of 196 flights with different velocities, payload weights and external conditions.

## Data availability

All data and materials used in the analysis are available at [UAV Attack Dataset] https://doi.org/10.21227/00dg-0d12, [ALFA Dataset] https://doi.org/10.1184/R1/12707963.v1, [IMCIS dataset] https://kaggle.com/competitions/icmcis-drone-tracking, [Package delivery UAV dataset] https://doi.org/10.1184/R1/12683453.v1, and [Ours] https://github.com/CKPerrusquia/CPhy-ML.git under an Apache 2.0 license for the purposes of reproducing and extending the analysis.

## Code availability

All code and materials used in the analysis ar available at https://github.com/CKPerrusquia/CPhy-ML.git under an Apache 2.0 license for the purposes of reproducing and extending the analysis (https://doi.org/10.5281/zenodo.10499878).

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

## Acknowledgements

This work was supported by the Engineering and Physical Sciences Research Council under Grant EP/V026763/1 and by the Royal Academy of Engineering and the Office of the Chief Science Adviser for National Security under the UK Intelligence Community Postdoctoral Research Fellowship programme.

## Author contributions

A.P., W.G., and B.F. conceptualized, proved theory, designed, performed research and analysed data. Z.W. contributed in the real-world experiment testing. A.P. and W.G. guided and supervised the work. All authors wrote the paper.

## Competing interests

The authors declare no competing interests.
