## [Peer Review File · Communications Engineering]

Reviewers' comments:

Reviewer #1 (Remarks to the Author):

Interesting research about physics–informed machine learning for drone control. The novel and interesting, but not-too original work is assembled into a decently drafted manuscript that needs some mild revisions.

- The manuscript is clear, relevant for the field and presented in a well-structured manner. The cited references are current (over eighty percent within the last 5 years), while suggested very recently published references are offered in review to aid revision since the broad field is neglected in favor of too quickly narrowly focusing on the references related to the proposed approaches. The manuscript is scientifically sound, and the experimental design is appropriate to test the hypothesis. The manuscript's results are reproducible based on the details given in the methods section. The figures/tables/images/schemes appropriate and properly show the data. They are easy to interpret and understand. The data is interpreted appropriately and consistently throughout the manuscript. The conclusions are consistent with the evidence and arguments presented.

The Abstract is okay but is not likely to entice the readership to continue reading the rest of the manuscript.

- Use of acronyms/abbreviations in an abstract is unlikely to attract readers not already aware of the manuscript's content. First–person tense is inappropriate.

- Results are only presented in a weak, qualitative fashion. Highest quality expression of main conclusions or interpretations is quantitative results discussed in the broadest context possible, e.g., percent performance improvement compared to a declared benchmark. "...We believe that this framework can provide deeper insight..." is very weakly stated results compared to "...xxx percent performance improvement over conventional methods was achieved...."

The Introduction is decently done with some omitted very recent literature and some mild abuse of multi-citation without elaboration (7 double-citations, 2 triple citations, 1 quadruple citations).

- Competing alternatives were neither qualitatively described in the literature review or quantitatively described in the manuscript, indicating a potential ignorance of the broader field that should be ameliorated in the revision.

- Please provide reasons for the readership to seek the multi citations offered without individualized justification for the reader to seek each reference.

- Originally developed for underwater drones, deterministic artificial intelligence was compared to physics–informed deep learning by Zhai in the already award–winning article <https://doi.org/10.3390/s22176362>.

- Alternatively, autonomous drone electronics amplified with Pontryagin–based optimization was proposed by Xu in *Electronics*, 2023.

Equations are scientifically sound and well presented, enhancing the manuscript quality.

Figures are decently done with some mandatory improvements to ensure the readership has access to the content.

- Internal font size is occasionally too small. Many parts of figure 1 are illegibly small. Abscissa and ordinant labels and legend texts are illegibly small in figures 3–5.

- Line styles and sizes are identical in figures 3,5,6 rendering the disparate data indistinguishable when the manuscript is read in printed hardcopy (particularly in black and white) negating the value of the

figures due to reliance on colors.

Tables are decently done to introduce problem formation (aiding repeatability), but quantitative results are neglected.

- Particularly for comparative figures (e.g., 5–6), please add a table of accompanying canonical figures of merit (e.g., means and deviations of difference, or others) to help the reader ascertain quantitative differences between the plotted data.
- For such a manuscript, heavy in acronym and variable usage, please add periodic tables of proximal definitions, so the readership is not required to flip back and forth between pages to remind themselves of acronym and variable definitions.
- Inclusion of a table defining variables and acronyms in an appendix is welcome and effective. Please add such.

Reviewer #2 (Remarks to the Author):

This work presents the CPhy-ML framework that uncovers two intention definitions related to trajectory and control objectives through robust inference capabilities, combining data-driven methods and physics-informed models. This framework is valuable for trajectory intention classification, employing a hybrid classifier and novelty detector, and for regression with expert models. It also uses reservoir computing for prediction, enhancing robustness and precision. Additionally, the CPhy-ML framework addresses the inference of reward functions for explaining drone decision-making, but further research is needed for reliability.

Some comments about style:

Few typos/mistakes, I recommend another round of proofreading:

Line 174: On On

It is odd to have the section "Methods" after the conclusions, why not to keep it as part of the methodology in section 3?

The Section on limitations would be better read as a subsection of the conclusion, perhaps a few not very long paragraphs summarising such limitations. Otherwise, I recommend a more comprehensive "Discussion" section, before the conclusions.

About the content:

I got confused as at the beginning of the paper the authors say: "Four trajectory intentions are used throughout the research: mapping flight, point-to-point flight, package delivery, and perimeter flight". But then, in the conclusions, it is stated that the proposed framework predicts two intention definitions related to trajectory and control objectives. Is it four or two?

My questions arose from the above were:

- How were these intentions chosen or who decided these intentions were relevant to be detected?
- How was it assessed that these trajectory intentions do not overlap or become a subset of one another, for instance, point-to-point flight could be a subset of perimeter flight.
- What is the operation frequency at which the trajectory is fed to the method/classifiers. That will have a huge impact on how the motion of the drone is being observed. Is it robust to variations in the frequency of the readings?
- Another important issue is noise, what happens with noise sensors? What are the limits when data becomes inaccurate.

I understand that simulation and dataset data were used to carry out this research, but it would have been desirable to show experiments with real flights as latency and frequency issues will have an impact on the methods, hence it is important to evaluate them under real conditions.

Finally, I strongly recommend the authors to reorganise to make it clearer and better ordered. Make sure the objectives are consistent with what is described in the experiment and conclusion sections. Properly justify what the four trajectory intentions were selected and show a scenario where a misbehaviour is really detected with your work, something I did not appreciate throughout the paper.

RESPONSE TO REVIEWERS

COMMS-23-0291-T: Uncovering Drone Intentions using
Control Physics Informed Machine Learning

Adolfo Perrusquía, Weisi Guo, Benjamin Fraser, Zhuangkun Wei

November 23, 2023

Reviewer 1

Please take into account that the equations numbers in the new version of the manuscript have changed. The equation numbers cited in the responses correspond to the new version of the manuscript. Interesting research about physics-informed machine learning for drone control. The novel and interesting, but not-too original work is assembled into a decently drafted manuscript that needs some mild revisions.

1. The Abstract is okay but is not likely to entice the readership to continue reading the rest of the manuscript. Use of acronyms/abbreviations in an abstract is unlikely to attract readers not already aware of the manuscript's content. First-person tense is inappropriate.

Thank you for your valuable comments to improve the quality and impact of the paper. In the revised version, we improve the abstract and avoid using acronyms and first-person tense throughout the complete manuscript. The changes in the abstract are the following:

- *First-time definition of the acronym CPhy-ML and avoid first-person tense:* Here, a novel control-physics informed machine learning (CPhy-ML) that can robustly infer across intention classes is developed.
 - *Performance improvement results:* The proposed CPhy-ML couples the representation power of deep learning with the conservation laws of aerospace models to reduce bias and instability. The proposed CPhy-ML achieves competitive state-of-the-art results in trajectory intention classification with an accuracy of 94.51-97.75%. On the other hand, 48.28% performance improvement over traditional regression and prediction methods was achieved across different time-windows and metrics. The reward inference results are notably improved over conventional inverse reinforcement learning approaches, decreasing the root mean squared spectral norm error from 3.3747 to 0.3229.
2. Results are only presented in a weak, qualitative fashion. Highest quality expression of main conclusions or interpretations is quantitative results discussed in the broadest context possible, e.g., percent performance improvement compared to a declared benchmark. "...We believe that this framework can provide deeper insight..." is very weakly stated results compared to "...xxx percent performance improvement over conventional methods was achieved..."

In the revised version, the statement "...We believe that this framework can provide deeper insight..." was removed from the abstract and changed with the assessment of quantitative results compared with the state-of-the-art. The improved abstract is given in page 1 of the manuscript as follows:

"Fully autonomous aerial platforms or drones are increasingly used across diverse application areas. Uncooperative drones do not announce their identity/flight plans and can pose a potential risk to critical infrastructures. Understanding drone's intention is important to assigning risk and executing countermeasures. Drones have rapidly flexible capabilities and diverse underpinning algorithms. This makes distinguishing malicious from naive intentions across platforms difficult. Intentions are often intangible and unobservable, and a variety of tangible intention classes are often inferred as a proxy. However, inference of drone intention classes using observational data alone is inherently unreliable due to observational and learning bias.

Here, a novel control-physics informed machine learning (CPhy-ML) that can robustly infer across intention classes is developed. The proposed CPhy-ML couples the representation power of deep learning with the conservation laws of aerospace models to reduce bias and instability. The proposed CPhy-ML achieves competitive state-of-the-art results in trajectory intention classification with an accuracy

of 94.51-97.75%. On the other hand, 48.28% performance improvement over traditional regression and prediction methods was achieved across different time-windows and metrics. The reward inference results are notably improved over conventional inverse reinforcement learning approaches, decreasing the root mean squared spectral norm error from 3.3747 to 0.3229.”

3. The Introduction is decently done with some omitted very recent literature and some mild abuse of multi-citation without elaboration (7 double-citations, 2 triple citations, 1 quadruple citations). Please provide reasons for the readership to seek the multi citations offered without individualized justification for the reader to seek each reference.

In the revised version, we reduce the multi-citation and provide more details of individual references. Double-citations are used when the references cover the same problem with slightly different methodologies.

4. Competing alternatives were neither qualitatively described in the literature review or quantitatively described in the manuscript, indicating a potential ignorance of the broader field that should be ameliorated in the revision.

The proposed CPhy-ML framework is quantitatively compared with other methods in the state-of-the-art. These methods cover a different deep neural model architectures evaluated across diverse metrics for both classification, regression and prediction. In the improved version, the proposed reward function inference algorithm is compared quantitatively with other state-of-the-art inverse reinforcement learning architectures. These comparative experiments are the ones that were missing in the manuscript. The following two paragraphs were added (see page 17-18 lines 393-405) in the manuscript to cover this gap:

“The performance of the proposed reward shaping IRL is compared against the gradient IRL [8] and model-based IRL [4] under diagonal and non-diagonal weight matrices. Knowledge of the exact weight matrix R is assumed for both the gradient and model-based IRL algorithms, with an initial weight matrix $Q_0 = 0_{n \times n}$. The gradient IRL uses a learning rate of $\gamma = 0.1$. For the reward-shaping IRL, random initial weight matrices are proposed.

Table 1: **Objective function inference results.** Root mean squared spectral norm error (RMSSNE) results across different weight matrices and IRL algorithms. Best results are in bold

e_i	RMSSNE					
	Gradient IRL		Model-Based IRL		Reward Shaping IRL (ours)	
	Diagonal weight matrices	Non-diagonal weight matrices	Diagonal weight matrices	Non-diagonal weight matrices	Diagonal weight matrices	Non-diagonal weight matrices
$\ K_{i+1} - K_i\ $	0.0096	0.0044	0.0072	0.0012	0.0273	0.0328
$\ P_{i+1} - P_i\ $	0.0061	0.0024	0.0136	0.0039	0.0641	0.0740
$\ Q_{i+1} - Q_i\ $	0.0562	0.0322	0.0486	0.0179	0.0907	0.1093
$\ R_{i+1} - R_i\ $	-	-	-	-	0.0197	0.0424
$\ K_i - K\ $	0.0480	0.0108	0.0153	0.0099	0.0649	0.0724
$\ P_i - P\ $	2.3886	6.5916	1.9761	3.3759	0.1331	0.1633
$\ Q_i - Q\ $	2.3910	6.2848	1.9802	3.3747	0.1942	0.3229
$\ R_i - R\ $	-	-	-	-	0.1051	0.2693

Table 1 summarizes the root mean squared spectral norm error (RMSSNE) results of the IRL algorithms under diagonal and non-diagonal weight matrices after 5,000 episodes. Despite the weight matrix R is assumed known for both the gradient and model-based IRL approaches, the results show they cannot converge to the real values. Furthermore, the initial weight matrix Q_0 plays a major role in the convergence of the respective IRL algorithm. On the other hand, the proposed reward-shaping IRL architecture overcomes these issues and simultaneously estimate both Q and R and verify the convergence to their near real values.”

- Originally developed for underwater drones, deterministic artificial intelligence was compared to physics-informed deep learning by Zhai in the already award-winning article 10.3390/s22176362. Alternatively, autonomous drone electronics amplified with Pontryagin-based optimization was proposed by Xu in Electronics, 2023.

Thank you for your suggestions. In the revised version we include the first reference [9] to motivate the use of physics informed models in line 50-53 of page 2 which is given in the next sentence

“Physics informed models have demonstrated improvements in the learning capabilities of data-driven methods [9]. These physics informed models appear either as a regularization term in the loss function [3] or from conservation laws [1] and a prior model structure [6].”

On the other hand, the suggested paper of autonomous drone electronics amplified with Pontryagin-based optimization was not used in the revised version since it does not address the intention inference problem.

Figure 1: Overview of the CPhy-ML for Drone Intention Inference.

- Figures are decently done with some mandatory improvements to ensure the readership has access to the content. Internal font size is occasionally too small. Many parts of figure 1 are illegibly small. Abscissa and ordinant labels and legend texts are illegibly small in figures 3-5.

In the revised version, Figure 1, 3-5 are improved based on your kindly suggestions. The new figures are given in Figs. 1-4 of this response letter.

- Line styles and sizes are identical in figures 3,5,6 rendering the disparate data indistinguishable

Figure 2: Trajectory Intention Regression example results.

Figure 3: Reservoir computing results of a single mission profile. Ground truth data is represented by a solid line, the linear RC results are depicted with a dotted line with triangle marker, and a dotted line represents the PIRC results.

when the manuscript is read in printed hardcopy (particularly in black and white) negating the value of the figures due to reliance on colors.

In the revised version, we modify the line style of Figs.3, 5 and 6 to make them clearer. This is shown in Fig. 2, Fig. 4 and Fig. 5 of this response letter.

8. Tables are decently done to introduce problem formation (aiding repeatability), but quantitative results are neglected.

Tables summarizes the quantitative results of the proposed CPhy-ML framework. Here, Tables 2, 3, 5, 6, 8-11, 14, 15, and 17 of the manuscript summarize the quantitative results of the proposed models compared with several state-of-the-art models. In addition, each table is explained accordingly to provide insights of the

Figure 4: **DMD linear model with LQR estimation.** The results with solid line corresponds to the raw measurements, while the results in dotted line stand for the DMD-LQR estimation.

Figure 5: **Reward function Inference Results.**

performance improvement using the proposed framework.

- Particularly for comparative figures (e.g., 5–6), please add a table of accompanying canonical figures of merit (e.g., means and deviations of difference, or others) to help the reader ascertain quantitative differences between the plotted data.

In the revised version, we add two additional tables that show the MSE and RMSSNE of the proposed DMD-LQR and model-based reward-shaping IRL algorithms. This permits to observe the benefits of the proposed models in comparison with other state-of-the-art algorithms. The new tables are given in Table 1 and Table 2 of this response letter (given in page 18 and 19 of the manuscript).

- For such a manuscript, heavy in acronym and variable usage, please add periodic tables of proximal definitions, so the readership is not required to flip back and forth between pages to remind themselves of acronym and variable definitions. Inclusion of a table defining variables and acronyms in an appendix is welcome and effective. Please add such.

We add periodic tables (see page 11, 13, and 15 of the manuscript) to facilitate readership. In addition, we add a table in the additional information section with the acronyms used throughout the manuscript. The new tables are given in Tables 3-6 of this response letter.

Table 2: MSE of the DMD-LQR algorithm across diverse periodic and non-periodic trajectories

MSE		
Position Axes	Periodic Trajectories	Non-Periodic Trajectories
X	0.06198	0.05709
Y	0.06277	0.05829
Z	0.06421	0.05947
Average	0.06299	0.05828

Table 3: Deep neural classifiers for comparisons.

Acronym	Description
LSTM	Long-short term memory
GRU	Gate Recurrent Unit
CBLSTM	Convolutional Bidirectional LSTM
CBLSTM	Convolutional Bidirectional LSTM with Attention
CNN	Convolutional Neural Network
CNNA	Convolutional Neural Network with Attention

Table 4: Deep neural regression models for comparisons

Acronym	Description
Multi-Input BLSTM	Multi-Input Bidirectional Long-Short Term Memory
Multi-Input CNN	Multi-Input Convolutional Neural Network
Multi-Input CBLSTMA	Multi-Input Convolutional BLSTM with Attention

Table 5: Reservoir computing (RC) models used for comparisons.

Model	Description
RC Linear	RC with Linear Decoder
RC SVM	RC with Support Vector Machine Decoder
RC MLP	RC with Multi-Layer Perceptron Decoder

Table 6: **Acronyms used throughout this research paper**

Acronym	Description
UAV	Unmanned Autonomous Vehicle
DL	Deep Learning
CPhy-ML	Control-Physics Informed Machine Learning
RF	Radio Frequency
EKF	Extended Kalman Filter
DNN	Deep Neural Network
RBF	Radial Basis Function
ReLU	Rectified Linear Unit
RNN	Recurrent Neural Network
LSTM	Long Short Term Memory
GRU	Gated Recurrent Unit
CBLSTM	Convolutional Bidirectional Long Short Term Memory
CBLSTMA	Convolutional Bidirectional Long Short Term Memory with Attention layer
CNN	Convolutional Neural Network
CNNA	Convolutional Neural Network with Attention
DMoE	Deep Mixture of Experts
RC Linear	Reservoir Computing with Linear Decoder
RC SVM	Reservoir Computing with Support Vector Machine Decoder
RC MLP	Reservoir Computing with Multi-Layer Perceptron Decoder
PIRC	Physics Informed Reservoir Computing
DMDc	Dynamic Mode Decomposition with Control.
LQR	Linear Quadratic Regulator
DMD-LQR	Dynamic Mode Decomposition with Linear Quadratic Regulator control
DARE	Discrete Algebraic Riccati Equation
ARE (CARE)	(Continuous) Algebraic Riccati Equation
IRL	Inverse Reinforcement Learning
MSE	Mean Squared Error
Recon MSE	Mean Squared Reconstruction Error
MAE	Mean Absolute Error
RMSE	Root Mean Squared Error
RMSSNE	Root Mean Squared Spectral Norm Error
R^2	Coefficient of determination

Reviewer 2

Please take into account that the equations numbers in the new version of the manuscript have changed. The equation numbers cited in the responses correspond to the new version of the manuscript.

1. Few typos/mistakes, I recommend another round of proofreading: Line 174: On On.

Thank you for your valuable comments to enhance the impact and clarity of the paper. In the revised version, we proofread the manuscript and correct all typos, grammatical errors and mistakes.

2. It is odd to have the section “Methods” after the conclusions, why not to keep it as part of the methodology in section 3?

In many Nature Publishing Group (NPG) journals, results before methods is often the publisher’s organisation, but we understand how this is counter intuitive for reviewers. In the revised version, we reorganize the sections of the manuscript to improve clarity and facilitate the readiness. Specifically, we move the methods section before the results section.

3. The Section on limitations would be better read as a subsection of the conclusion, perhaps a few not very long paragraphs summarising such limitations. Otherwise, I recommend a more comprehensive “Discussion” section, before the conclusions.

In the revised version, the limitations are given as a subsection of the conclusions section. Here, we try to explain in detail the limitations of the approach to clearly state future research vectors that can benefit this research area. This will create an harmonious research community that deals with the challenges in counter drone systems.

4. I got confused as at the beginning of the paper the authors say: “Four trajectory intentions are used throughout the research: mapping flight, point-to-point flight, package delivery, and perimeter flight”. But then, in the conclusions, it is stated that the proposed framework predicts two intention definitions related to trajectory and control objectives. Is it four or two?

Thank you for noticing this. In the revised version, we provide two intention definitions given by: trajectory intention and reward intention. Each definition of intention possesses their own intention classes. For the trajectory intention, we use four classes: package delivery, surveillance, point-to-point and perimeter flight. For the reward intention, two classes are defined normal and anomalous trajectories. This is stated in page 2 and 3 of the manuscript as follows:

“The proposed model deals with two different but complementary definitions of intention: trajectory intention and reward function intention. Trajectory intention is associated to the purpose of use of the drone and the potential trajectory profile that the drone will follow in future time steps. The reward function intention describes the hidden motivation used for the control design; this scalar function is the one that the user wants to optimize in an infinite horizon to accomplish any desired task. Four trajectory intention classes are used throughout the research based on the available open-access datasets which cover: mapping, point-to-point, package delivery, and perimeter flights. In addition, two reward function classes are used: normal and anomalous trajectories. These reward intention classes are determined based on the inferred reward function which weights each mixture of state and control input trajectories to achieve a desired behaviour.”

5. How were these intentions chosen or who decided these intentions were relevant to be detected?
- How was it assessed that these trajectory intentions do not overlap or become a subset of one another, for instance, point-to-point flight could be a subset of perimeter flight.

In the revised version, four trajectory intention classes are used based on the available open-access datasets which cover: mapping, point-to-point, package delivery, and perimeter flights. These intention classes are described as follows: i) mapping flights represent flights where a particular region of interest is mapped from images to form a large top-down representation of the region, ii) point-to-point flights cover long-term transit flights following a straight line between two distant waypoints, iii) package delivery represents flights from real-world package delivery flight experiments, and iv) perimeter flights include flights that the starting and ending location point is the same, that is, they followed a closed-loop perimeter pattern. We define the trajectory intention classes which are independent according to these previous definitions.

6. What is the operation frequency at which the trajectory is fed to the method/classifiers. That will have a huge impact on how the motion of the drone is being observed. Is it robust to variations in the frequency of the readings?

In the revised version, the trajectories obtained from open-access datasets are pre-processed by either upsampling or downsampling them to 1 Hz. These trajectories are converted into sub-trajectory features according to different time-window sizes. We used window-sizes of 8, 16, 32 and 64 that define the amount of data used to feed the neural models and verify their robustness. The trajectories obtained from personal use drone are obtained from the VICON camera system with a frequency of 120 Hz. For these data, the only requirement is to exhibit enough richness to either estimate a drone’s linear model or infer the reward function. This is stated in page 4 lines 134-138 of the manuscript:

“A custom radar simulation process is developed based on the Stone Soup software [2] and the open-access telemetry data discussed at Datasets. These datasets are preprocessed as follows: 1) converting latitude, longitude, and altitude into local Cartesian coordinates; 2) removing unwanted periods (e.g., take-off, on ground); and 3) down/up-sampling to 1 Hz. It is assumed that each measurement of the simulated radar has Gaussian noise.”

In page 5 lines 150-153:

“The trajectory tracks are processed into several sub-trajectories each with an associated intention label. This is to cover the needs of real-time prediction using partial information of the complete trajectory and observe the robustness of the deep models. Four different window sizes are used in this research: 8, 16, 32 and 64 seconds.”

And in page 14 line 323-327 of the manuscript:

“A personal drone is used to conduct real-world testing. The drone comprises a beagle-bone-blue (BBB) chip as its central processor, T-1045 frames and propellers, KV-8816 motors with compatible electronic speed controllers, and a 4-cell 14.8V-5000 mAh LiPo battery. The VICON camera system, composed of 25 well-distributed cameras with different resolutions, is used to track the position of the drone. The VICON measurements and transmitting frequency is 120 Hz with a localization error of 0.01-0.5 meter.”

7. Another important issue is noise, what happens with noise sensors? What are the limits when data becomes inaccurate.

In the revised version, all the data used in this paper is assumed to contain Gaussian noise. This noise is attenuated using extended Kalman filter (EKF) for the open-access datasets. This is stated in page 5 lines 138-141 of the manuscript as follows:

“An extended Kalman Filter (EKF) is used to obtain the final simulated radar

tracks based on the following nonlinear model per axis

$$\begin{aligned} \mathbf{x}_t &= \mathbf{F}_t \mathbf{x}_{t-1} + \boldsymbol{\omega}_t, \quad \boldsymbol{\omega}_t \sim \mathcal{N}(\mathbf{0}, \mathbf{Q}_t) \\ \mathbf{x}_t &= \begin{bmatrix} x_{\text{pos}} \\ x_{\text{vel}} \end{bmatrix}, \quad \mathbf{F}_t = \begin{bmatrix} 1 & dt \\ 0 & 1 \end{bmatrix}, \quad \mathbf{Q}_t = \begin{bmatrix} \frac{dt^3}{3} & \frac{dt^2}{2} \\ \frac{dt^2}{2} & dt \end{bmatrix} q. \end{aligned} \quad (1)$$

where x_{pos} and x_{vel} are the Cartesian position and velocity of the x -axis, q is the velocity noise diffusion constant which is set to 0.1 to obtain smooth track estimations that closely matches with the original flight trajectories.”

Dynamic mode decomposition (DMD) is used for the personal-use drone data to obtain a linear model that gives noise-free trajectories. The DMD-LQR results are described in page 16-17 of the manuscript as follows:

“To this end, first, a dynamic mode decomposition with control (DMDc) [5] model is applied to the RF data to obtain a linear model that preserves the dynamic modes of the real non-linear drone dynamics. It is used the Euler angles: roll ϕ , pitch θ , and yaw ψ ; and the total thrust force μ as control inputs. Using these measurements as control inputs allows to construct a simple discrete linear model. It is observed that the richness of the trajectory is crucial to ensure a good generalization of the model, e.g., point-to-point trajectories are not useful since the dynamic modes of the drone are not excited [7]. In addition, due to the high non-linear dynamics of the drone, then different linear matrices are obtained for different trajectories despite of being from the same drone. To alleviate this problem, the trajectories that exhibit more richness are used to generate the linear model.

Figure 6: **DMD linear model with LQR estimation.** The results with solid line corresponds to the raw measurements, while the results in dotted line stand for the DMD-LQR estimation. The first row shows the estimated trajectories of the trained linear model. The second row shows the Euler angles control inputs obtained from the LQR design. The third row shows the generalization capabilities of the estimated linear model under different trajectories

Fig. 6 shows the estimated trajectories of the drone’s data under the DMDc linear model in closed-loop with an user-design LQR controller. One of the main advantages of this approach is that the nonlinear physics of the drone is transformed into a linear system. This transformation facilitates the prediction analysis with noise suppression. Here, the closed-loop system between the DMDc linear model and

the LQR controller is able to track different trajectories accurately and satisfies the small angle condition for the Euler angles control input. Table 7 summarizes the MSE results across all the telemetry data trajectories obtained from custom flights (see RF). The results show the estimated linear system under the LQR control is capable to estimate accurately both periodic and non-periodic trajectories. Moreover, the inferred states are noise-free which is a requirement in most machine learning techniques to avoid biased predictions. Here, the proposed DMD-LQR approach can be regarded as an effective tool for noise-suppression.

Table 7: MSE of the DMD-LQR algorithm across diverse periodic and non-periodic trajectories

Position Axes	MSE	
	Periodic Trajectories	Non-Periodic Trajectories
X	0.06198	0.05709
Y	0.06277	0.05829
Z	0.06421	0.05947
Average	0.06299	0.05828

8. I understand that simulation and dataset data were used to carry out this research, but it would have been desirable to show experiments with real flights as latency and frequency issues will have an impact on the methods, hence it is important to evaluate them under real conditions.

In the revised version, we use a personal-use drone to capture diverse trajectories with control input to assess some elements of the proposed framework. This is stated in page 14 lines 323-327 of the manuscript as follows:

“A personal drone is used to conduct real-world testing. The drone comprises a beagle-bone-blue (BBB) chip as its central processor, T-1045 frames and propellers, KV-8816 motors with compatible electronic speed controllers, and a 4-cell 14.8V-5000 mAh LiPo battery. The VICON camera system, composed of 25 well-distributed cameras with different resolutions, is used to track the position of the drone. The VICON measurements and transmitting frequency is 120 Hz with a localization error of 0.01-0.5 meter.”

9. Finally, I strongly recommend the authors to reorganise to make it clearer and better ordered. Make sure the objectives are consistent with what is described in the experiment and conclusion sections. Properly justify what the four trajectory intentions were selected and show a scenario where a misbehaviour is really detected with your work, something I did not appreciate throughout the paper.

In the revised version, we reorganize the paper according to your valuable comments and clearly motivate the objectives of each part of the proposed framework. We provide two definitions of intention given by trajectory intention and reward intention. Here, the four trajectory intention classes (surveillance, package delivery, point-to-point, perimeter flight) belong to the trajectory intention definition, other trajectories that do not exhibit the profile of the proposed trajectory classes are assumed to be anomalous and require further investigation. On the other hand, reward intention has two classes based on the values of the reward function which determine if the drone poses a normal or anomalous behaviour. Anomalous trajectories are detected when the mean squared reconstruction error is high. Furthermore, anomalous trajectories are identified by means of the reward function values. Here, high reward values imply that the drone is operating outside the desired behaviour.

References

- [1] Peter J Baddoo, Benjamin Herrmann, Beverley J McKeon, J Nathan Kutz, and Steven L Brunton. Physics-informed dynamic mode decomposition. *Proceedings of the Royal Society A*, 479(2271):20220576, 2023.
- [2] David Last, Paul Thomas, Steven Hiscocks, Jordi Barr, David Kirkland, Mamoon Rashid, Sang Bin Li, and Lyudmil Vladimirov. Stone soup: announcement of beta release of an open-source framework for tracking and state estimation. In *Signal Processing, Sensor/Information Fusion, and Target Recognition XXVIII*, volume 11018, pages 52–63. SPIE, 2019.
- [3] Christian Legaard, Thomas Schranz, Gerald Schweiger, Ján Drgoňa, Basak Falay, Cláudio Gomes, Alexandros Iosifidis, Mahdi Abkar, and Peter Larsen. Constructing neural network based models for simulating dynamical systems. *ACM Computing Surveys*, 55(11):1–34, 2023.
- [4] Bosen Lian, Yusuf Kartal, Frank L Lewis, Dariusz G Mikulski, Gregory R Hudak, Yan Wan, and Ali Davoudi. Anomaly detection and correction of optimizing autonomous systems with inverse reinforcement learning. *IEEE Transactions on Cybernetics*, 2022.
- [5] Abhinav Narasingam and Joseph Sang-Il Kwon. Development of local dynamic mode decomposition with control: Application to model predictive control of hydraulic fracturing. *Computers & Chemical Engineering*, 106:501–511, 2017.
- [6] Adolfo Perrusquía and Weisi Guo. Physics informed trajectory inference of a class of nonlinear systems using a closed-loop output error technique. *IEEE Transactions on Systems, Man, and Cybernetics: Systems*, 2023.
- [7] Ee Weinan. A proposal on machine learning via dynamical systems. *Communications in Mathematics and Statistics*, 1(5):1–11, 2017.
- [8] Wenqian Xue, Patrik Kolaric, Jialu Fan, Bosen Lian, Tianyou Chai, and Frank L Lewis. Inverse reinforcement learning in tracking control based on inverse optimal control. *IEEE Transactions on Cybernetics*, 52(10):10570–10581, 2021.
- [9] Hanfeng Zhai and Timothy Sands. Comparison of deep learning and deterministic algorithms for control modeling. *Sensors*, 22(17):6362, 2022.

REVIEWERS' COMMENTS:

Reviewer #1 (Remarks to the Author):

Thanks for essentially ceded all the revisions requested. The manuscript is very much improved. I would recommend publication after amelioration of figure 6's display of the convergence of the elements of Q_{i+1} and R_{i+1} in Θ_i to their approximately exact values Q and R , respectively. The attempt to simultaneously display all the information on two very small plots using identical line sizes and styles is ineffective, rendering the figures a complete waste. The information should be displayed on multiple, larger plots using varied line styles and thickness or markers to insure the readership can discern the information, particularly when the proposed archival document is printed in black and white many years from now.

Reviewer #2 (Remarks to the Author):

The authors have attended my concerns. Thank you.

RESPONSE TO REVIEWERS

COMMS-23-0291-T: Uncovering Drone Intentions using
Control Physics Informed Machine Learning

Adolfo Perrusquía, Weisi Guo, Benjamin Fraser, Zhuangkun Wei

January 13, 2024

Reviewer 1

1. Thanks for essentially ceded all the revisions requested. The manuscript is very much improved. I would recommend publication after amelioration of figure 6's display of the convergence of the elements of Q_{i+1} and R_{i+1} in Θ_i to their approximately exact values Q and R , respectively. The attempt to simultaneously display all the information on two very small plots using identical line sizes and styles is ineffective, rendering the figures a complete waste. The information should be displayed on multiple, larger plots using varied line styles and thickness or markers to insure the readership can discern the information, particularly when the proposed archival document is printed in black and white many years from now.

Thank you for your valuable comments to improve the quality and impact of the paper. We consider your comment and improve the quality of Figure 6 as follows

Figure 1: **Convergence results of the Reward function Inference Algorithm.** The first column shows the results for diagonal weight matrices and the second one the results for non-diagonal weight matrices. The first row shows the convergence results of the spectral norm error of the kernel matrix P_i in blue and orange lines with square and cross markers. The second row shows the convergence results of the control gain K_i in green and yellow lines with circle and triangle markers. The third row shows the spectral norm error of the reward weight matrix Q_i with pink and red lines with right triangle and plus markers. The fourth row shows the convergence results of the weight R_i with black and gray lines with left triangle and diamond markers. The fifth row shows the convergence of the elements of Q_{i+1} and R_{i+1} in Θ_i to their approximately exact values Q and R , respectively.

Reviewer 2

1. The authors have attended my concerns. Thank you.

Thank you for your comments that clearly enhances the impact of our paper. Thank you!